**1** **A 1-km daily surface soil moisture dataset of enhanced coverage**

**2** **under all-weather conditions over China in 2003-2019**

**3** Peilin Song[1,4†], Yongqiang Zhang[1]*, Jianping Guo[2]*, Jiancheng Shi[3], Tianjie
**4** Zhao[4], Bing Tong[2]

**5** [1] Key Laboratory of Water Cycle and Related Land Surface Processes, Institute of Geographic Sciences

**6** and Natural Resources Research, The Chinese Academy of Sciences, Beijing 100101, China

**7** [2] State Key Laboratory of Severe Weather, Chinese Academy of Meteorological Sciences, Beijing

**8** 100081, China

**9** [3] National Space Science Center, Chinese Academy of Sciences, Beijing 100190, China

**10** [4] State Key Laboratory of Remote Sensing Science, Aerospace Information Research Institute, Chinese

**11** Academy of Sciences. Beijing 100101, China

**12** [†] now at School of Electronic Science and Engineering, Xi'an Jiaotong University, Xi'an, 710049, China

**14** *Correspondence to*: Yongqiang Zhang (zhangyq@igsnrr.ac.cn); Jianping Guo ( jpguo@cma.gov.cn )

## **Abstract:**

Surface soil moisture (SSM) is crucial for understanding the hydrological process of our earth surface. Passive microwave (PM) technique has long been the primary tool for estimating global SSM from the view of satellite, while the coarse resolution (usually >~10 km) of PM observations hampers its applications at finer scales. Although quantitative studies have been proposed for downscaling satellite PM-based SSM, very few products have been available to public that meet the qualification of 1-km resolution and daily revisit cycles under all-weather conditions. In this study, we developed one such SSM product in China with all these characteristics. The product was generated through downscaling the AMSR-E/AMSR-2 based SSM at 36-km, covering all on-orbit time of the two radiometers during 2003-2019. MODIS optical reflectance data and daily thermal infrared land surface temperature (LST) that had been gap-filled for cloudy conditions were the primary data inputs of the downscaling model, so that the "all-weather" quality was achieved for the 1-km SSM. Daily images from this developed SSM product have quasi-complete coverage over the country during April-September. For other months, the national coverage percentage of the developed product is also greatly improved against the original daily PM observations, through a specifically developed sub-model for filling the gap between seams of neighboring PM swaths during the downscaling procedure. The product is well compared against *in situ* soil moisture measurements from 2000+ meteorological stations, indicated by station averages of the unbiased RMSD ranging from 0.052

vol/vol to 0.059 vol/vol. Moreover, the evaluation results also show that the developed product outperforms the SMAP-Sentinel (Active-Passive microwave) combined SSM product at 1-km, with a correlation coefficient of 0.55 achieved against that of 0.40 for the latter product. This indicates the new product has great potential to be used for hydrological community, agricultural industry, water resource and environment management.

# 1. Introduction

Surface soil moisture (SSM) is one of the most important variables that dominate the mass and energy cycles of earth surface system (Entekhabi et al., 2010b). Satellite-based SSM datasets of sufficiently fine spatio-temporal resolutions over large-scale areas have significant implication on improved investigations at various research fields including hydrological signature identification (Zhou et al., 2021; Jung et al., 2010), agricultural yield production estimation (Ines et al., 2013; Pan et al., 2019), drought/waterlogging monitoring and warning (Vergopolan et al., 2021; Den Besten et al., 2021; Jing and Zhang, 2010), as well as weather prediction and future climate analysis (Koster et al., 2010; Jeffrey et al., 2001). Microwave bands with centimeter-level or longer wavelengths (X-band, C-band, and L-band) are currently identified as the primary band channels suitable for SSM observations from view of satellite, due to their high penetration capabilities through cloud layers and vegetation canopies. In terms of sensor types, microwave SSM detection includes passive microwave (radiometer-based) techniques and active microwave (radar, scatterometer) techniques.

**64** Satellite-based passive microwave (PM) radiometers, e.g. the Soil Moisture Active

**65** Passive (SMAP), the Soil Moisture and Ocean Salinity (SMOS), and the Advance

**66** Microwave Scanning Radiometer-2 (AMSR-2), can obtain SSM observations at a

**67** revisit interval of 1-3 days, with relatively poor native spatial resolutions of tens of

**68** kilometers. Active microwave (AM) such as radar can achieve kilometer-level and even

**69** finer resolution of observations targeting at the earth surface. However, this usually

**70** sacrifices the swath width of radar configuration, because of which, most satellite-based

**71** synthetic aperture radars (SAR) have an obviously longer global revisit cycle (usually

**72** longer than 5 days, e.g. Sentinel-1 SAR data) than the typical radiometers. Moreover,

**73** AM radar backscatter signals are extremely sensitive to speckle noise (Entekhabi et al.,

**74** 2016), as well as influence from soil roughness, vegetation canopy structure and water

**75** content (Piles et al., 2009). All above influential factors have seriously impeded the use

**76** of AM radar techniques or combination of passive/active microwave datasets for

**77** producing high spatial resolution SSM products with a frequent revisit.

**78** Apart from microwave signals, solar reflectance or ground emission signals

**79** originated from optical and infrared band domains also have the potential to reflect

**80** SSM variation. Based on optical/infrared bands, however, SSM is typically estimated

**81** based on indirect relationships through intermediate variables like soil evaporation

**82** (Komatsu, 2003), vegetation condition (Zeng et al., 2004), or soil thermal inertia

**83** (Verstraeten et al., 2006). To overcome the spatio-temporally instable performance on

**84** SSM modelling that might be brought by such indirect relationships, they are typically

**85** fused with the PM SSM datasets. In this manner, it can well reconcile the advantage of

PM observations with respect to its high sensitivity to SSM variation, as well as that of
optical/infrared observations with respect to its finer spatial resolutions at kilometer- or
even hectometer-levels. Such data fusion techniques are also known as downscaling
techniques of PM remote sensing SSM. Archetypal downscaling models include the
"universal triangle feature space (UTFS)"-based models (Chauhan et al., 2003; Choi
and Hur, 2012; Sanchez-Ruiz et al., 2014), the "DISaggregation based on a Physical
And Theoretical scale CHange (DISPACTH)" model (Merlin et al., 2010; Merlin et al.,
2005; Merlin et al., 2013; Merlin et al., 2008), and the "University of California, Los
Angeles (UCLA)" model (Peng et al., 2016). The physics of these models are mainly
based on the response of SSM variation to changes in soil evaporation or land surface
evapotranspiration. Another significant branch of such downscaling models are based
on the sensitivity of SSM to soil thermal inertia, which is quantified by diurnal LST
difference estimated from thermal-infrared wave bands (Fang and Lakshmi, 2013; Fang
et al., 2018).
Sabaghy et al. (2020) have shown that using optical and infrared data can achieve
finer-resolution SSM estimates which are better consistent with ground soil moisture
records, compared with using the radar datasets. Moreover, considering the short revisit
cycle (daily) of optical/infrared sensors onboard typical polar-orbit satellites, using
optical/infrared datasets to downscale PM SSM should be among the optimal methods
for obtaining SSM data with high spatio-temporal resolutions over national, continental,
or global scales. On the other hand, satellite remote sensing SSM products that are
characterized by 1-km resolution of daily revisit intervals and stable long time series

**108** dating back to at least 15-20 years ago, are urgently required for accelerating the

**109** development of various research fields, especially agriculture industry, water resources

**110** management, and hydrological disaster monitoring (Sabaghy et al., 2020; Mendoza et

**111** al., 2016). However, very seldom sets of such data products are publicly available to

**112** the remote sensing research community because of the following drawbacks. First,

**113** there is a serious lack of cloud-free optical/infrared imagery, which means the method

**114** cannot deliver any SSM downscaling under cloudy/rainy weather. Second, most of the

**115** above-mentioned optical/infrared-data-based downscaling methods were mainly

**116** evaluated at regional or even smaller scales. This might raise concern on the

**117** universality of those methods. For example, the DISPATCH method has been

**118** recognized to be less effective in humid (energy-limited) regions compared with in arid

**119** and semi-arid (water-limited) regions (Molero et al., 2016; Song et al., 2021; Zheng et

**120** al., 2021). As far as the UTFS-based method is concerned, a poorer performance was

**121** obtained compared to the DISPATCH in a typical water-limited region in North

**122** America, according to the experiment conducted by Kim and Hogue (2012).

**123**    To improve the above-mentioned issues, we produced an all-weather daily SSM

**124** data product at 1-km resolution all over China during 2003-2019, based on fusion of

**125** multiple remote sensing techniques, including reconstruction of optical/infrared

**126** observations under cloud as well as an improved PM SSM downscaling methodology

**127** proposed in our previous study (Song et al., 2021). The potential significance of this

**128** study includes

(i) to better serve and investigate the land surface hydrology processes and their
sophisticated interactions to human society at multi-scale (from national to regional)
resolutions in China because the country covers about 1/15 of the global terrestrial area
with about 1/5 of the world population, and
(ii) to provide a methodology framework that can inspire future studies on
generating similar SSM datasets all over the globe, based on the plentifulness of
resources on climate type, land covers, and topography in China.

## 137   2. Methods and Materials

## 138   2.1 Datasets

2.1.1 PM SSM data
Spatial downscaling of PM SSM is the fundamental theory for constructing the
target finer-resolution data product in this study. Therefore, the native retrieval
accuracy of the coarse-resolution PM SSM data product, based on which the
downscaling procedures are performed, is considerably crucial to the performance of
the downscaled data product (Busch et al., 2012; Im et al., 2016; Kim and Hogue, 2012).
Although the L-band PM brightness temperature (TB) observed by satellite missions
such as SMAP or SMOS are considered more suitable for SSM retrieval compared with
C- or X-band TB, both above missions started their space operations in the 2010s. This
means that to obtain downscaled SSM of longer historical periods, we still require to
rely on other C-/X-band-based radiometers which started their operations earlier than
SMAP and SMOS. An optimal satellite PM TB observation system dating back to
earlier years of this century is composed of the "Advanced Microwave Scanning
Radiometer of the Earth Observing System (AMSR-E)", together with its successor of
AMSR-2. AMSR-E operated during 2002-2011 onboard the Aqua satellite which is
governed by National Aeronautics and Space Administration (NASA), whilst AMSR-
2 is operating onboard the Global Change Observation Mission1-Water (GCOM-W1)
satellite developed by the Japan Aerospace Exploration Agency (JAXA) since 2012.
Several classical PM SSM retrieval algorithms have been applied to the afore-
mentioned "AMSR series (including AMSR-E and AMSR-2)" TB for generating long-
term global SSM products at 25 km (Table 1), including the JAXA algorithm (Fujii et
al., 2009; Koike et al., 2004), the "Land Parameter Retrieval Model (LPRM)" algorithm
(Song et al., 2019b; Meesters et al., 2005; Owe et al., 2001), and the algorithm
developed by the University of Montana (UMT) (Jones et al., 2009; Du et al., 2016). A
recent AMSR-based night-time SSM product during 2002-2019 has been produced
through a neural network trained against SMAP radiometer-based descending SSM
(hereafter referred to as "NN-SM product") (Yao et al., 2021). The global validation
results show that this NN-SM product is better than the JAXA and LPRM products.
Besides, the NN-SM has also been compared with another long-term ~25-km all-
weather SSM dataset generated through the European Space Agency (ESA)'s Climate
Change Initiative (CCI) program. The ESA-CCI SSM product is different from the rest
products mentioned above in that it was implemented by fusion of observations from
comprehensive AM- and PM-based satellite sensors, rather than only relying on the
radiometers of AMSR series. According to Yao et al. (2021), the ESA-CCI SSM has
slightly better validation accuracy than the NN-SM product, but the number of available
observations per pixel cell in an entire year is much smaller for the ESA-CCI SSM in
Southeast China. In view of all above coarse-resolution SSM data products, we finally
selected the NN-SM product to implement the following spatial downscaling
procedures rather than the ESA-CCI SSM, to make a balance between data accuracy
and data availability per year. We have also made additional evaluations within China
in Section Appendix-A to ensure the relatively outstanding performance of the NN-SM
product as described above.

Table 1 Information of all-weather microwave remote sensing coarse-resolution SSM data

products that can be potentially downscaled to obtain fine resolution SSM.

| Name | Resolution | Satellite radiometers involved | Data availability (URL) |
|---|---|---|---|
| NN-SM product | 36 km (by the EASE Grid projection) | AMSR-E/ AMSR-2 (2002-2011, 2012-present) | https://data.tpdc.ac.cn/en/data/c26201fc-526c-465d-bae7-5f02fa49d738/ |
| ESA-CCI v6.1 product | 0.25° | AMSR-E/ AMSR-2/ SMOS/ WindSat/ SMMR/ SSM/I/ TMI (1978-2020) | https://www.esa-soilmoisture-cci.org/v06.1_release |
| JAXA product | 0.25° / 0.1° | AMSR-E/ AMSR-2 (2002-2011, 2012-present) | https://gportal.jaxa.jp/ |
| LPRM product | 0.25° / 0.1° | AMSR-E/ AMSR-2 (2002-2011, 2012-present) | https://search.earthdata.nasa.gov/ |
| UMT product | 25 km (by the EASE Grid projection) | AMSR-E/ AMSR-2 (2002-present) | http://files.ntsg.umt.edu/data/LPDR_v2/ |

**183**

**184**    2.1.2 Optical remote sensing data and digital elevation model (DEM)

**185**    Optical remote sensing datasets provide finer spatial texture information on the

**186**    daily basis for the downscaling purpose of PM SSM. Such data that can be used as

**187**    inputs of our SSM product processing line are mainly provided by the Moderate-

**188**    resolution Imaging Spectroradiometer (MODIS) onboard the Terra and Aqua satellites.

**189**    Specifically, they involve the 1-km daily night-time Aqua MODIS LST product

**190**    (MYD21A1N.v061) and the 500-m daily "Bidirectional Reflectance Distribution

**191**    Function (BRDF)" - Adjusted Reflectance dataset (MCD43A4.v061). MYD21A1 LST

**192**    data can be recognized as a crucial proxy of land surface thermal capacity (Fang et al.,

**193**    2013) and soil evaporative rate (Merlin et al., 2008). The MCD43A4 nadir reflectance

**194**    product, with view angle effect corrected using the BRDF model, is capable to provide

**195**    observations from visible to shortwave-infrared bands that can characterize water

**196**    content variation of the bare soils as well as the vegetation canopy. Overall, the above-

**197**    mentioned datasets were selected primarily because they deliver indicators (land

**198**    surface thermal capacity, soil evaporative rate, or vegetation condition) that can well

**199**    response to soil moisture dynamics from different aspects. Prior to being employed for

**200**    SSM downscaling, conventional pre-processing procedure of pixel quality check was

**201**    applied for both optical products by screening out pixels not classed as "good quality",

**202**    according to the 8-bit "Quality Assessment (QA)" field of each spectral band. Moreover,

**203**    to normalize their natively different spatial resolutions, all MCD43A4 based reflectance

**204** values at the 500-m pixel level were upscaled to the sinusoidally projected MODIS 1-

**205** km grids using their spatial averages.

**206**     Apart from MODIS optical remote sensing data, all 90-m DEM tiles generated by

**207** the NASA Shuttle Radar Topography Mission (SRTM; http://srtm.csi.cgiar.org/, last

**208** access: July 10, 2020) were mosaicked all over China and then employed as another

**209** essential input variable for the procedures as described by Section 2.2.2 below. Similar

**210** to that applied to the MCD43A4 product, spatial upscaling in correspondence to the

**211** MODIS 1-km grids is also an indispensable pre-processing step for the mosaicked

**212** DEM data.

**213** 2.1.3 Study area and validation data

**214**     Our study area is set up as the total terrestrial extent of China. To comprehensively

**215** evaluate the SSM downscaling performances for different geographic regions (see

**216** Section 3.3), we divided the country further into six different geographic-climate

**217** regions using elevation, precipitation, hydrogeology, vegetation type, and topography.

**218** The six regions include the Northeast Monsoon (NEM) region, the Northwest Arid

**219** (NWA) region, the Qinghai–Tibet Plateau (QTP) region, the North China Monsoon

**220** (NCM) region, the South China Monsoon (SCM) region, and the Southwest Humid

**221** (SWH) region. The detailed delimitation principle of these geographic-climate regions

**222** was originally described in Meng et al. (2021). The geographic zoning map is shown

**223** in Fig. 1, while the corresponding shapefile boundary files can be accessed from the

**224** Resource and Environment Science and Data Center of the Chinese Academy of

**225** Sciences (http://www.resdc.cn/, last access: May 22, 2021).

**226**

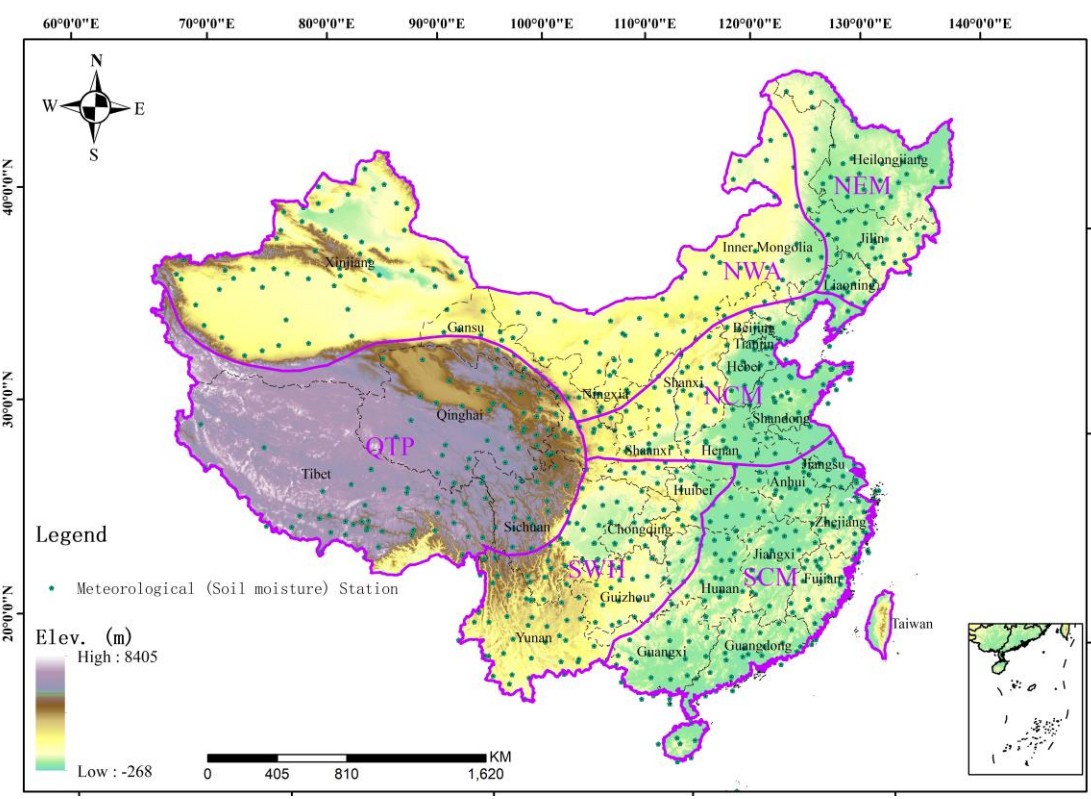

**228** Fig. 1 The geographic zoning map of China (delineated using the purple color) superposed with

**229** topographic information, as well as general locations for the 756 basic meteorological stations

**230** (http://data.cma.cn/, last access: January 20, 2021) that provide partial benchmark measurements for

**231** SSM and LST validation in this study.

**232** We utilized ground soil moisture measurements for validating the downscaled

**233** remote sensing SSM product at the local scale. The ground measurements are derived

**234** from 2417 meteorological stations (including 756 basic stations of the National Climate

**235** Observatory and 1661 regionally intensified stations) of over China, as partially shown

**236** in Fig. 1. The soil moisture measurement devices in these stations, with uniform

**237** observation standards, are instrumented under the national project of "Operation

Monitoring System of Automatic Soil Moisture Observation Network in China (Wu et
al., 2014  )", the construction of which has been led by China Meteorological
Administration since 2005. Until 2016, all stations have been in operation for
automatically observing hourly in situ soil moisture dynamics at eight different depth
ranges (0-10 cm, 10-20 cm, 20-30 cm, 30-40 cm, 40-50 cm, 50-60 cm, 70-80 cm, 90-
100 cm). It has also been widely used by previous studies for evaluating satellite soil
moisture estimates in China (Meng et al., 2021; Chen et al., 2020; Zhang et al., 2014;
Zhu and Shi, 2014). In our current study, ground measurements matching the shallowest
depth range (0-10 cm) from the initial time of each station until the end of 2019 are
employed as validation benchmark of the satellite SSM retrievals. At the temporal
dimension, measurements made at 1:00 A.M. and 2:00 A.M are averaged, in order to
match the mean satellite transit time of 1:30 A.M. for AMSR descending observations.
Moreover, 0-cm top ground temperatures are simultaneously measured at all these
meteorological stations on the daily basis, at the local time windows of 2:00 A.M./P.M.
and 10:00 A.M./P.M., respectively. We therefore exploited such measurements
recorded at 2:00 A.M. to validate the cloud gap-filled night-time (~1:30 A.M.) LST
estimates over the Aqua-MODIS based 1-km pixels containing these stations (see
Section 2.2.2). Our primary validation period covers the entire years of 2017, 2018, and

2019.

In addition to the ground soil moisture measurements, the SMAP Level3
radiometer-based          daily          36-km          SSM          product
(https://dx.doi.org/10.5067/OMHVSRGFX38O  ) in its descending orbit scenes (at

**260** ~6:00 A.M. of local time) from 2016 to 2019, was employed as another complemental

**261** validation benchmark. This dataset is potential for providing more comprehensive

**262** evaluations to our developed product at regional/national scales, especially on account

**263** of its notably creditable performance (see Fig. A1 in Appendix A). The latest version

**264** of this dataset (SPL3SMP, Version 8) contains soil moisture retrievals based on

**265** different algorithms including the dual channel algorithm and the single channel

**266** algorithm. In this study we only extracted SSM estimates derived with the dual channel

**267** algorithm because this algorithm was reported to outperform the single channel

**268** algorithm over some agricultural cropland core validation sites (O'neill et al., 2021).

**269** 2.1.4 Ancillary SSM products for comparison

**270** In order to comprehensively demonstrate the validation performance of our

**271** proposed SSM product, there is necessity to make an inter-comparison against similar

**272** existing datasets. In this regard, we introduced the Level2 SMAP/Sentinel Active-

**273** Passive combined SSM product on 1-km earth-fixed grids, i.e., the SPL2SMAP_S_V3

**274** dataset (Das et al., 2020), and used its validation performance against in-situ

**275** measurements throughout the years of 2017, 2018, and 2019, as a baseline to better

**276** evaluate our proposed SSM product. The SPL2SMAP_S_V3 dataset contains global

**277** SSM at resolutions of 3 km and 1 km respectively, which were disaggregated from the

**278** SMAP radiometer-based SSM retrievals of 36-km/9-km footprints in conjunction with

**279** the high-resolution Sentinel-1 C-band radar backscatter coefficients (Das et al., 2019).

**280** To our knowledge, this dataset is possibly the only publicly available product which

**281**   can provide global remote sensing SSM estimates at the 1-km resolution. The sentinel

**282**   backscatter coefficient inputs for this product are only those received in the descending

**283**   orbit scenes (at ~6:00 A.M. of local time), whilst the closest SMAP SSM retrievals

**284**   from either ascending (at ~6:00 P.M. of local time) or descending orbits are used to

**285**   spatially match up with the sentinel-1 scene. It is noticed that at the descending

**286**   observation time the soil moisture vertical profile has approached a hydrostatic balance

**287**   (Montaldo et al., 2001), thereby providing the optimal chance for soil moisture fusion

**288**   and validation with observations at different soil depths. Therefore, we only selected

**289**   the 1-km disaggregated SSM estimates based on descending SMAP SSM retrievals (i.e.,

**290**   the subset with field name of 'disagg_soil_moisture_1 km' in the SPL2SMAP_S_V3

**291**   dataset). Meanwhile, the 0-10 cm in-situ soil moisture measurements observed at 6:00

**292**   A.M. and the SMAP radiometer-based descending SSM estimates were employed as

**293**   the validation benchmarks, in a manner similar to that applied to our proposed SSM

**294**   product (Section 2.1.3).

**295**   ## 2.2   Methodology

**296**   The general methodological framework for producing the all-weather daily 1-km

**297**   SSM product is shown as in Fig. 2, with details described in the following context of

**298**   this section.

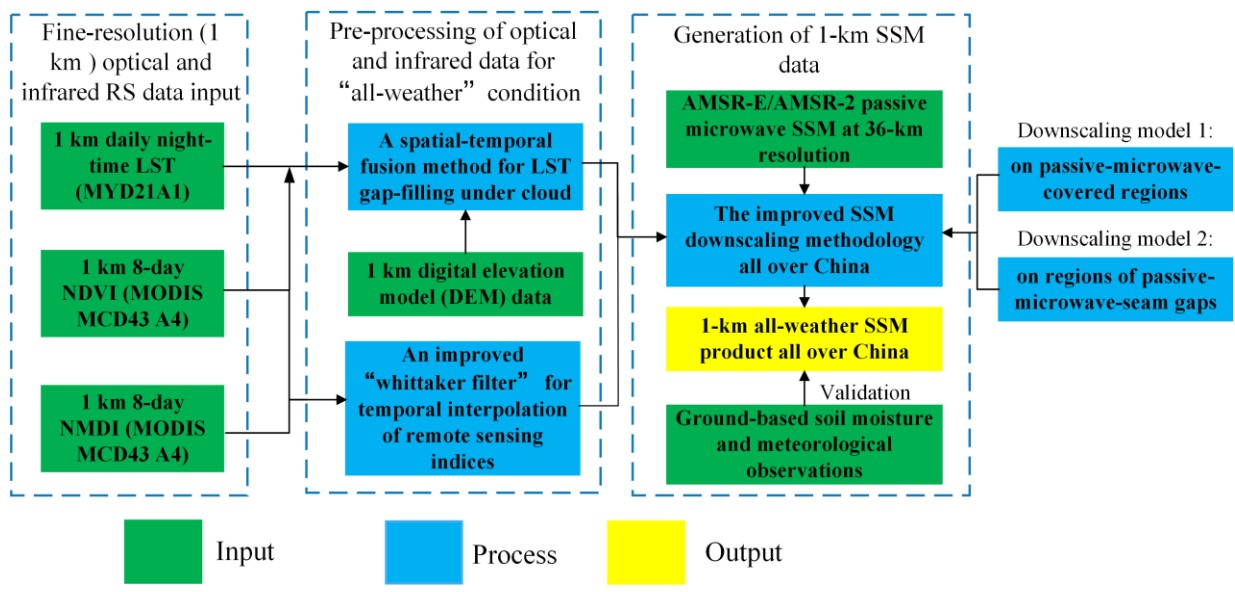

 Fig. 2 The overall methodological framework of this study.

**301** 2.2.1    Reconstruction of thermal-infrared LST and remote sensing (vegetation)

**302** indices under cloud

**303**     Reconstruction of missing pixels under cloud in the optical remote sensing input

**304** datasets is the prerequisite for achieving the "all-weather" property of the final

**305** downscaled SSM output. For reconstructing thermal-infrared LST, we adopted the

**306** cloud gap-filling method as proposed by our previous study (Song et al., 2019a). This

**307** method, also referred to as a typical "spatio-temporal data fusion" (STDF) method

**308** (Dowling et al., 2021), was built using clear-sky LST observations of spatially

**309** neighboring pixels observed at proximal dates, with concurrent NDVI and DEM also

**310** employed as additional data inputs. The STDF method can be expressed as follows:

**311**
$$LST^{*}_{t_1} = a \times LST^{*}_{t_0} + b \times NDVI^{*}_{t_1} + c \times DEM^{*} + d \qquad (1)$$

**312** Where the superscript "*" indicates that this variable has been normalized to the range

**313** 0 to 1.0 (Song et al., 2019a), based on the maximum and minimum values of that

**314** variable found across China (excluding invalid values representing states of snow, ice,

**315** and water bodies). Parameters *a, b, c*, and *d* are coefficients fitted between all pixels

**316** with clear-sky LST estimates on a specific date $t_1$ ($LST^{*}_{t1}$) and their counterparts on

**317**     one proximal date, $t_0$ ($LST^*_{t0}$). $NDVI^*_{t1}$ indicates the corresponding (normalized) NDVI

**318**     on the $t_1$ date calculated using the MCD43A4 daily product. After deriving the

**319**     coefficients of *a, b, c,* and *d*, Equation (1) was used to fill all cloudy MODIS LST pixels

**320**     on the $t_1$ date. For any $t_1$ date included in the study period, the $t_0$ date was iterated among

**321**     all neighboring dates of $t_1$ meeting the condition $| t_0 - t_1 | <= 30$ (from the nearest date to

**322**     the furthest date). The average of estimated LST values for $t_0$ was then taken where a

**323**     cloud gap pixel was filled more than once (based on the iterative $t_0$ dates). The iteration

**324**     was stopped when the fraction of pixels with effective LST values on $t_1$ was equal to or

**325**     exceeded 0.99.

**326**     An important flaw of this STDF method should be noticed with regard to

**327**     potentially existential bias of the cloud gap-filled LST outputs, because the outputs

**328**     represent theoretically reconstructed LST under clear sky rather than under the real

**329**     cloudy condition. Another of our previous studies (Dowling et al., 2021) concerning

**330**     this STDF method proposed a follow-up step, which incorporated PM-derived surface

**331**     temperature, to adjust that bias. In our current production pipeline, however, this

**332**     follow-up step for cloud bias adjustment in LST was not carried out. This is because

**333**     the results in Section Appendix-B show that using LST generated by the STDF alone

**334**     leads to more accurate SSM outcomes in general. The possible reasons for this are

**335**     discussed in Section 4.2.

**336**     Reconstruction of the remote sensing vegetation indices under cloudy conditions,

**337**     including NDVI and MNDI, was simply based on the modified time series filter of the

**338**     Whiitaker Smoother (MWS) as developed by Kong et al. (2019). This is reasonable

**339**     because the dynamic trends of vegetation growth are relatively less volatile compared

**340**     to LST on the daily basis, and can thus be gap-filled for missing values using a time-

**341**     series-filtering-like algorithm.

**342** 2.2.2   Improved downscaling technique of SSM based on fusion of PM and

**343** optical/infrared data

**344**      The core component of the SSM downscaling methodology is an improved linking

**345** model between PM SSM and (fine-resolution) optical remote sensing observations.

**346** This model enhances the relatively poorer performance of the conventional DISPATCH

**347** in energy-limited regions, whilst maintains the generally good quality of the

**348** DISPATCH in water-limited ones. Therefore, the improved model is more appropriate

**349** to be applied in China which contains a wide range of geographical settings, compared

**350** to other conventional downscaling models.  Since this model origins from our previous

**351** study (Song et al., 2021), herein we simply give its mathematical expression as follows:

**352**
$$SSM = \frac{a \times \ln(1 - SEE)}{1 - b \times NMDI} + c \tag{2}$$

**353** In Equation (2), *SEE* denotes "soil evaporative efficiency" and is a mathematical

**354** function of LST and the typical Normalized Difference Vegetation Index (NDVI), with

**355** its specific form described in Merlin et al. (2008). NMDI is another remote sensing

**356** index   calculated   as   $\frac{R_{infr,860nm} - (R_{sw,1600nm} - R_{sw,2100nm})}{R_{infr,860nm} + (R_{sw,1600nm} - R_{sw,2100nm})}$   (Wang   and   Qu,   2007).

**357** $R_{infr,860nm}$, $R_{infr,1600nm}$ , and $R_{infr,2100nm}$ represent land surface reflectance signals

**358** derived  from  three  different  MODIS-MCD43A4  based  near-infrared/shortwave-

**359** infrared bands, with their wavelengths centering at 860 nm, 1600 nm, and 2100 nm

**360** respectively. The parameters *a, b,* and *c* are empirical coefficients that represent

**361** background information of local soil texture and vegetation types. In Song et al. (2021),

**362** these coefficients have been fitted and calibrated based on multi-temporal observations

**363** at the PM pixel scale. In our current study, however, we have discovered that coupling

**364** of multiphase observations at both the spatial and the temporal dimensions can lead to

**365** more optimal solution of the coefficients, as they can produce downscaled SSM images

**366** with notably declined effect of 'mosaic' against the original PM 36-km pixels.

**367** Therefore, the modified optimal cost function $\chi^2$ for deriving these coefficients is re-

**368** defined as follows:

**369**
$$\chi^2 = \sum_{d=-dl}^{dl} \sum_{i=0}^{N=ws \times ws} w_i \times (SSM_{ob,i,d} - SSM_{mod,i,d})^2 \tag{3}$$

**370** Through the cost function, the spatial extent of each 36-km pixel $P_0$ on any arbitrary

**371** date $D_0$ obtains a unique set of coefficients. As shown by Equation (3), all pixels were

**372** exploited within the spatial square window (with its side length equal to $ws$) centered

**373** at $P_0$ ranging from $-dl$-th day to $dl$-$th$ day relative to the date of $D_0$. To determine the

**374** optimum values for $dl$ and $ws$, we have tested each member in the collection of [3, 5, 7,

**375** 9, 11, 13] for both parameters. Evaluation against in-situ data indicates that the

**376** optimum $dl$ and $ws$ are 5 and 7, respectively (results are similar to what is shown in

**377** Section 3.2, but not presented here). $SSM_{ob}$ and $SSM_{mod}$ denote the AMSR NN-SM 36-

**378** km SSM observations as well as SSM observations modelled by Equation (2) based on

**379** upscaled optical datasets, respectively. $w_i$ is a weight coefficient used to ensure that

**380** neighboring observations near the centering pixel $P_0$ play more dominating roles as

**381** compared with the far-end pixels in the cost function, considering the "Tobler's First

**382** Law of Geography (Sui, 2004)" . $w_i$ is calculated using an adaptive bi-square function:

**383**
$$w_i = [1-(\frac{dis_i}{b})^2]^2, dis_i < b$$
$$w_i = 0, dis_i >= b \tag{4}$$

where $dis_i$ indicates the distance between the i-*th* pixel and the centering pixel $P_0$. *b* is
named as the adaptive kernel bandwidth of the bi-square function (Duan and Li, 2016),
and is optimized as 200 km through using a cross validation method as recommended
by Brunsdon et al. (1996).
With the linking model obtained, we can subsequently utilize the spatial
downscaling relationship function to produce 1-km fine resolution SSM. The
downscaling relationship function is constructed by transforming the linking model into
its Taylor expansion formula and preserving all components with respect to the input
optical variables of the linking model at first and second orders. This relationship is
inspired from Malbéteau et al. (2016) and Merlin et al. (2010), and is mathematically
described below:
$SSM_{1\text{-}km} = SSM_{36km} + (\frac{\partial SSM}{\partial SEE})_{36km} \times (SSE_{1km} - <SSE>_{36km}) + 0.5 \times (\frac{\partial^2 SSM}{\partial SEE^2}) \times (SSE_{1km} -$
$<SSE>_{36km})^2 + (\frac{\partial SSM}{\partial NMDI})_{36km} \times (NMDI_{1km} - <NMDI>_{36km}) + 0.5 \times (\frac{\partial^2 SSM}{\partial NMDI^2}) \times$
$(NMDI_{1km} - <NMDI>_{36km})^2$ (5)
In the above relationship, $<>$ denotes the spatial averaging operator for all of the 1-km
optical remote sensing input variables within the corresponding 36-km pixel,
$\frac{\partial SSM}{\partial SEE} (\frac{\partial^2 SSM}{\partial SEE^2})$ and $\frac{\partial SSM}{\partial NMDI} (\frac{\partial^2 SSM}{\partial NMDI^2})$ respectively denoting the first-(second-) order
partial derivative of the linking model described in Equation (2).
It should be noticed that there exist middle-/low-latitude gap regions between
seams of neighboring daily AMSR-E(-2) swaths, indicating that $SSM_{36km}$ in Equation
(5) is not always available on the daily basis (Song and Zhang, 2021a). For such PM-
seam gaps on a particular date $t_0$, the corresponding $SSM_{36km,t0}$ in Equation (5) is
substituted by $0.5 \times (SSM_{36km,t0+1} + SSM_{36km,t0-1})+ \Delta SSM_{36km,t0}$. Herein $SSM_{36km,t0-1}$
and $SSM_{36km,t0+1}$ respectively denote the SSM estimate before and after the date of $t_0$.
$\Delta SSM_{36km,t0}$ is a component for correcting inter-day bias, with the following expression:

$$
\begin{aligned}
\Delta SSM_{36km,t0} = {} & SSM\left(SEE_{36km,t0},\ NMDI_{36km,t0}\right) - \\
& 0.5 \times \left(SSM\left(SEE_{36km,t0-1},\ NMDI_{36km,t0-1}\right) + SSM\left(SEE_{36km,t0+1},\ NMDI_{36km,t0+1}\right)\right)
\end{aligned}
\tag{6}
$$

In the above equation, $SSM(SEE_{36km}, NMDI_{36km})$ denotes SSM that is directly
modelled based on Equation (1) using 36-km SEE and NMDI. The 36-km SEE and
NMDI are obtained via averaging the variables spatially from their native resolution at
1-km. If all $SSM_{36\text{-}km}$ during the three consecutive days ($t_0$-1, $t_0$, and $t_0$+1) are missing
due to other extreme conditions like snow, ice, or surface dominated by substantially
large water bodies, the downscaling process cannot be fulfilled and all 1-km sub-pixels
with the $SSM_{36\text{-}km}$ have to be set as null values.
2.2.3    Evaluation metrics
We employed the classic metrics of 'Root Mean Square Difference (RMSD)' and
correlation coefficient (*r*-value) for evaluating satellite-based (SSM and LST) estimates
against ground measurements. Herein RMSD is not referred to as 'Root Mean Square
Error (RMSE)', although the latter term shares the same definition and has been used
more commonly in previous studies. This is because both ground observations and
other benchmark data (i.e. SMAP radiometer-based SSM) may also present
measurement uncertainties in practice. For SSM evaluation, the unbiased RMSD, or
ubRMSD (Entekhabi et al., 2010a; Molero et al., 2016), is calculated instead of RMSD
when validated against ground soil moisture measurements. This can better investigate
the time series similarity between satellite and in situ datasets by eliminating the
systematic bias caused by spatial scale mismatch between them.
The above-mentioned classic metrics are primarily suitable to evaluate the
absolute reliability of an independent remote sensing product. However, we also require
another metric for characterizing the relative improvement of the downscaled SSM
estimates against the original PM observations on capturing local soil moisture
dynamics. For this purpose, we employed the "gain metric" of $G_{down}$, which was
developed particularly by Merlin et al. (2015) for assessment of soil moisture
downscaling methodology. $G_{down}$ is a comprehensive indicator for evaluating gains of
the downscaled SSM against the original coarse-resolution PM data in terms of their
mean bias, bias in variance (slope), and time series correlation with ground benchmark.
It has a valid domain between -1 and 1, with positive (negative) value indicating
improved (deteriorated) spatial representativeness of the downscaled SSM against the
original PM data. Detailed definition and introduction of $G_{down}$ are given in Equation
(8) and Section 3.3 of Merlin et al. (2015).
## 3. Results
## 3.1   Evaluation on reconstructed thermal-infrared LST under
cloud
The meteorological-station-based validation of reconstructed 1-km thermal-
infrared LST under cloud were preliminarily fulfilled, to ensure the high quality of input
dataset variables for SSM downscaling. Since disadvantageous effects might be

**448** brought to this validation campaign by the potentially existing heterogeneity of the

**449** validated 1-km thermal-infrared remote sensing pixels, we firstly analyzed correlations

**450** between estimated and benchmark datasets at each station, only based on satellite

**451** remote sensing observations obtained under clear sky. Stations that have their

**452** correlation coefficients ($r_{clr}$) lower than 0.9 herein have to be screened out because there

**453** exist higher chances of cross-scale spatial mismatch within and around these stations

**454** in terms of the land surface thermal properties. Among all 2417 stations (see Section

**455** 2.1.3) where 0-cm in-situ top-ground temperature measurements were available, we

**456** finally preserved 2107 stations characterized by $r_{clr}$ >0.9. In the subsequent step, remote

**457** sensing LST under cloud and under clear-sky conditions were respectively validated at

**458** these stations, with the results revealed in Fig. 3. It is manifested through Fig. 3-(a) and

**459** -(b) that very close performances have been achieved between the clear-sky and the

**460** cloudy scenarios, especially considering their almost equally high validating

**461** correlations between 0.94-0.96. For each independent station, we calculated the

**462** "RMSD difference (RMSD_diff)" between the two scenarios, based on the formula of

**463** "$RMSD_{clr}$- $RMSD_{cld}$ (the subscripts of *'clr' and 'cld'* denote clear-sky and cloudy

**464** conditions separately)". The statistical distribution of this RMSD difference with regard

**465** to different stations is shown in Fig. 3-(c). Apparently, 1942 stations all over the country

**466** have obtained an RMSD difference value below 2.6 K, and the mean RMSD difference

**467** is about 1.9 K. All above results have indicated that the uncertainty of our night-time

**468** LST reconstruction algorithm proposed for cloudy conditions is not very significant.

**469**     The corresponsive uncertainty that could be propagated to downscaled SSM in this

**470**     stage is analyzed below in Section 3.2.

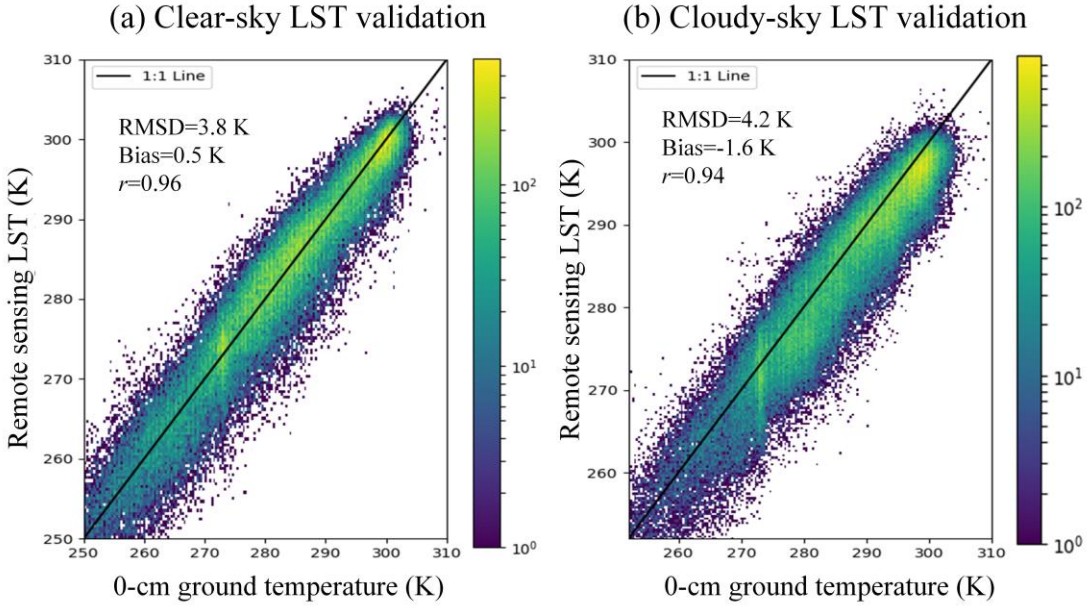

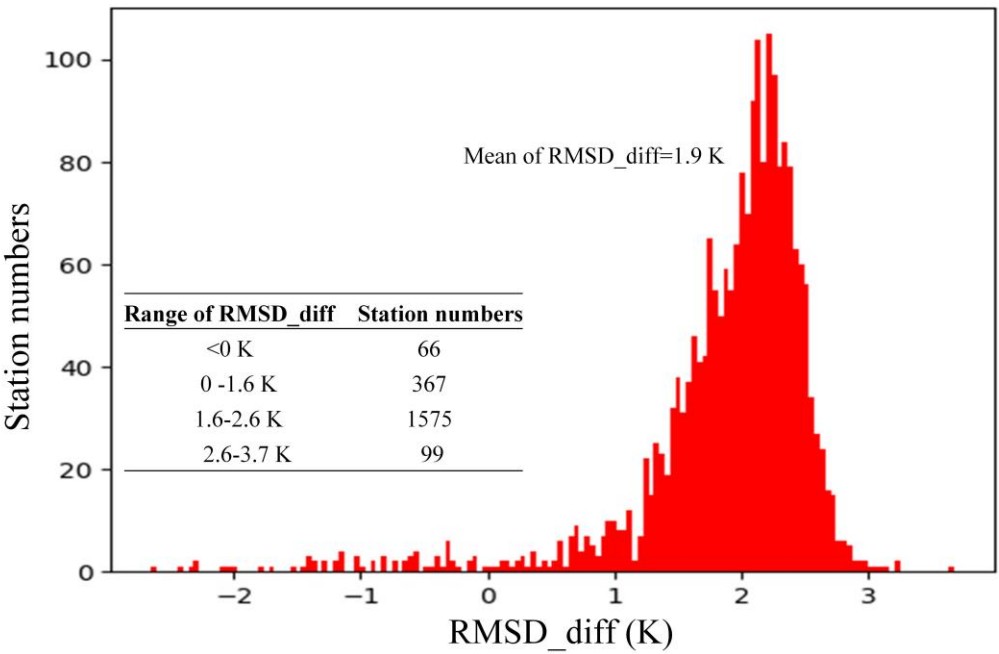

**471**

**472**     Fig. 3 validation results of the cloud gap-filled LST in China. (a) Density plot of thermal infrared

**473**     LST under clear-sky condition compared to the 0-cm ground temperature measurements for all

**474**     stations. (b) Same to (a) but for thermal infrared LST under cloudy conditions. (c) Statistical

distribution of difference between RMSD of clear-sky LST and RMSD of gap-filled LST under cloudy

condition with regard to different meteorological stations over the study region.


## 478  3.2 Evaluation on the final 1-km SSM product

The overall validation results of the finally downscaled 1-km SSM product against
ground soil moisture data is demonstrated in Fig. 4.  Fig. 4-(a) shows that about 85%
(N: 1833) of the total 2154 stations (the remaining 263 stations are located in pixels
with no effective PM observations and are thus removed) have obtained significantly
positive downscaling gains ($G_{down}$>0.03). This hints that the 1-km SSM product can
better capture the dynamic behaviors of local ground soil moisture data than the original
36-km PM NN-SM data, revealing higher spatial representativeness of the downscaled
SSM data product over the country. According to Fig. 4-(b), the mean ubRMSD of all
stations is about 0.054 vol/vol, while 90% of those stations have the number lower than
0.088 vol/vol. In addition, we made another analysis concerning the possible influence
of land cover types on SSM downscaling performance in Fig. 4-(c). The spatial
information of land cover types was derived from the MODIS MCD12Q1
(10.5067/MODIS/MCD12Q1.006) IGBP-based land use image in 2019. For stations
that experienced land use change throughout the years of the study period, the ubRMSD
is only reported for data in the year of 2019. Clearly, better accuracies are observed
mainly in grassland, cropland and bare soil surface, whilst relatively poorer
performances (with averages of ubRMSD higher than 0.06 vol/vol) are seen in urban
regions, (woody) savanna, and crop-to-natural-vegetation mosaic areas. Such a relative

**497**    performance across land covers is logical because all the land cover types with their

**498**    average ubRMSD higher than 0.06 vol/vol are characterized by lower hydrologic

**499**    homogeneity in terms of their definition, e.g. savanna, which is a mixture of grass and

**500**    tall trees, and urban areas, which are composed of impervious underlying surface.

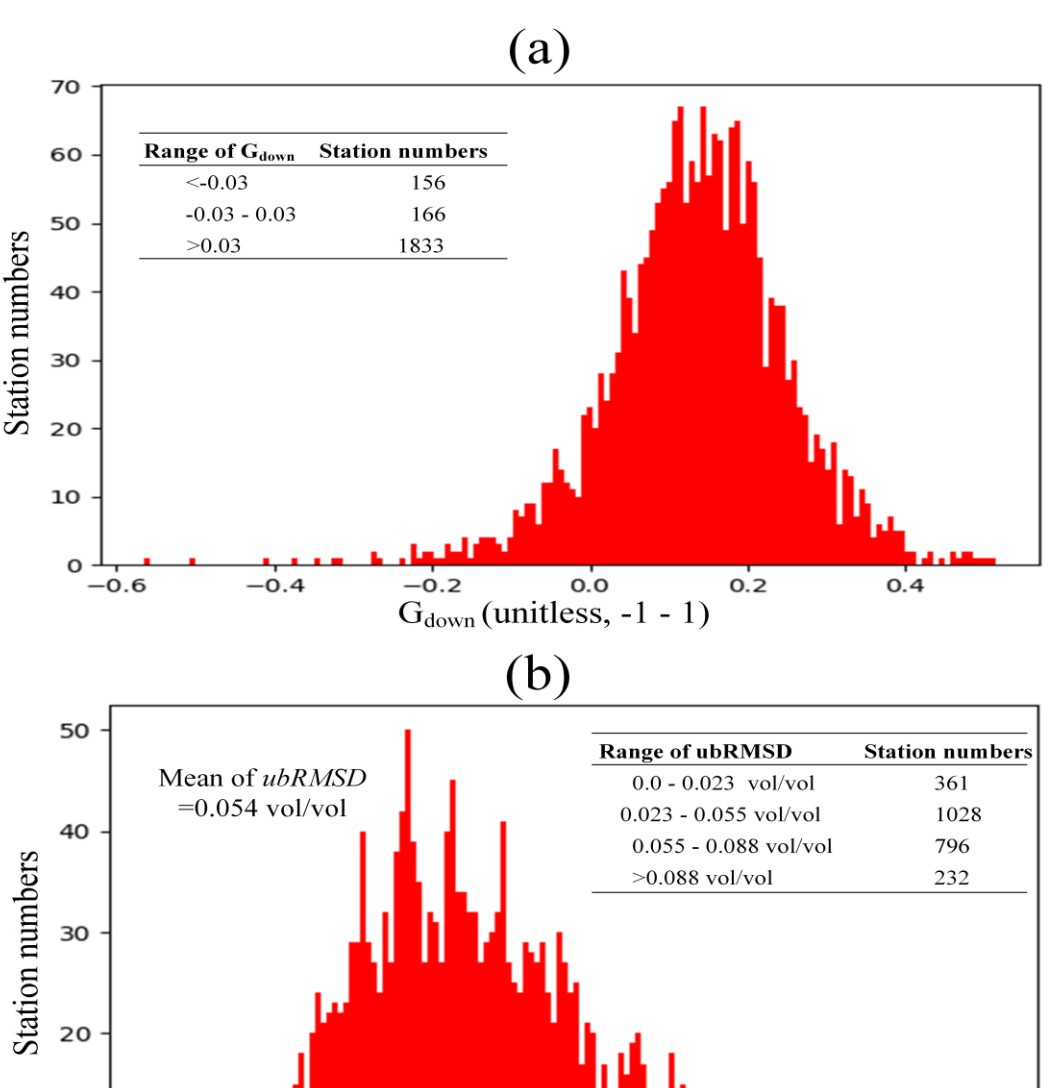

(a)

| Range of $G_{down}$ | Station numbers |
| --- | --- |
| <-0.03 | 156 |
| -0.03 - 0.03 | 166 |
| >0.03 | 1833 |

$G_{down}$ (unitless, -1 - 1)

(b)

Mean of *ubRMSD* =0.054 vol/vol

| Range of ubRMSD | Station numbers |
| --- | --- |
| 0.0 - 0.023  vol/vol | 361 |
| 0.023 - 0.055 vol/vol | 1028 |
| 0.055 - 0.088 vol/vol | 796 |
| >0.088 vol/vol | 232 |

ubRMSD (vol/vol)

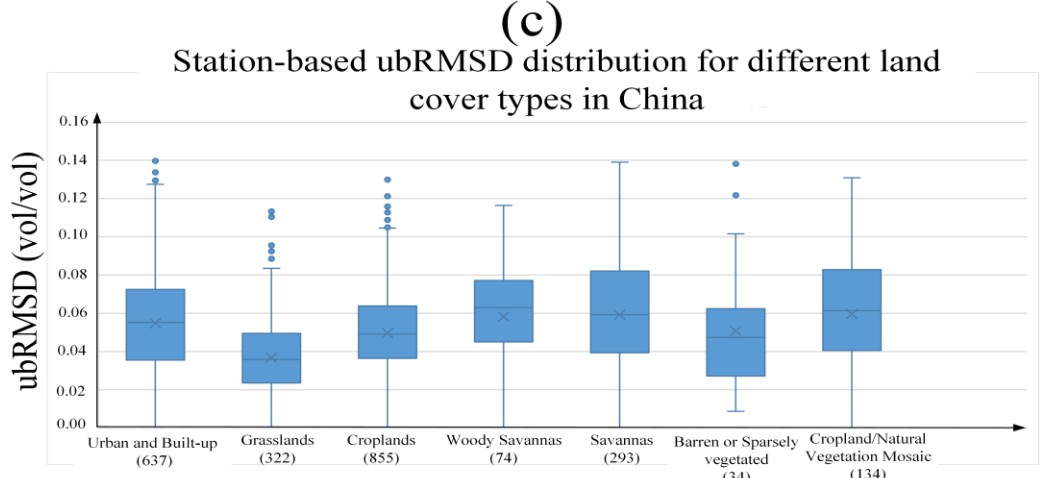

(c)

Station-based ubRMSD distribution for different land cover types in China

Land Cover Type

Fig. 4 General validation results of the currently developed SSM product. (a) $G_{down}$ distribution for
different stations over China. (b) ubRMSD distribution for different stations over China. (c) ubRMSD
statistics reported for different land covers. The numbers in the parentheses of the x-axis labels
represent the amount of meteorological stations corresponding to that specific land cover type.
In Fig. 5, we compared time series of regionally aggregated SSM from our
developed 1-km SSM product to that from the SMAP 36-km descending SSM, for each
of the six different geographic-climate regions (as shown in Fig. 1) from 2016 to 2019.
Via this effort, we mainly aim to reveal the consistency degree on reflecting soil
moisture temporal dynamics at different geographical settings between the two SSM
products. This also provides another view to evaluate the reliability of our developed
product. Because the SMAP radiometer has a slightly longer revisit cycle (~2-3 days)
than AMSR-2, the time series data are also aggregated and averaged at the temporal
dimension, with a displayed revisit cycle equal to three days. Overall, the time series
data correlate well with each other for all six regions. The relatively lower RMSDs
(<0.02 vol/vol) are found in regions with comparatively sparser vegetation covers
including the NWA region, the QTP region, and the NCM region. For other three dense-
vegetation regions, the performances of our developed product are slightly poorer. This
is especially the case for the SCM region, with a lower $r$-value of 0.84. The reason can
be attributed to the enlarged difference on penetration depth into the soil layers between
L-band (SMAP) and C-/X-/K- band (AMSR-2) emissions under dense vegetation
covers (Ulaby and Wilson, 1985).

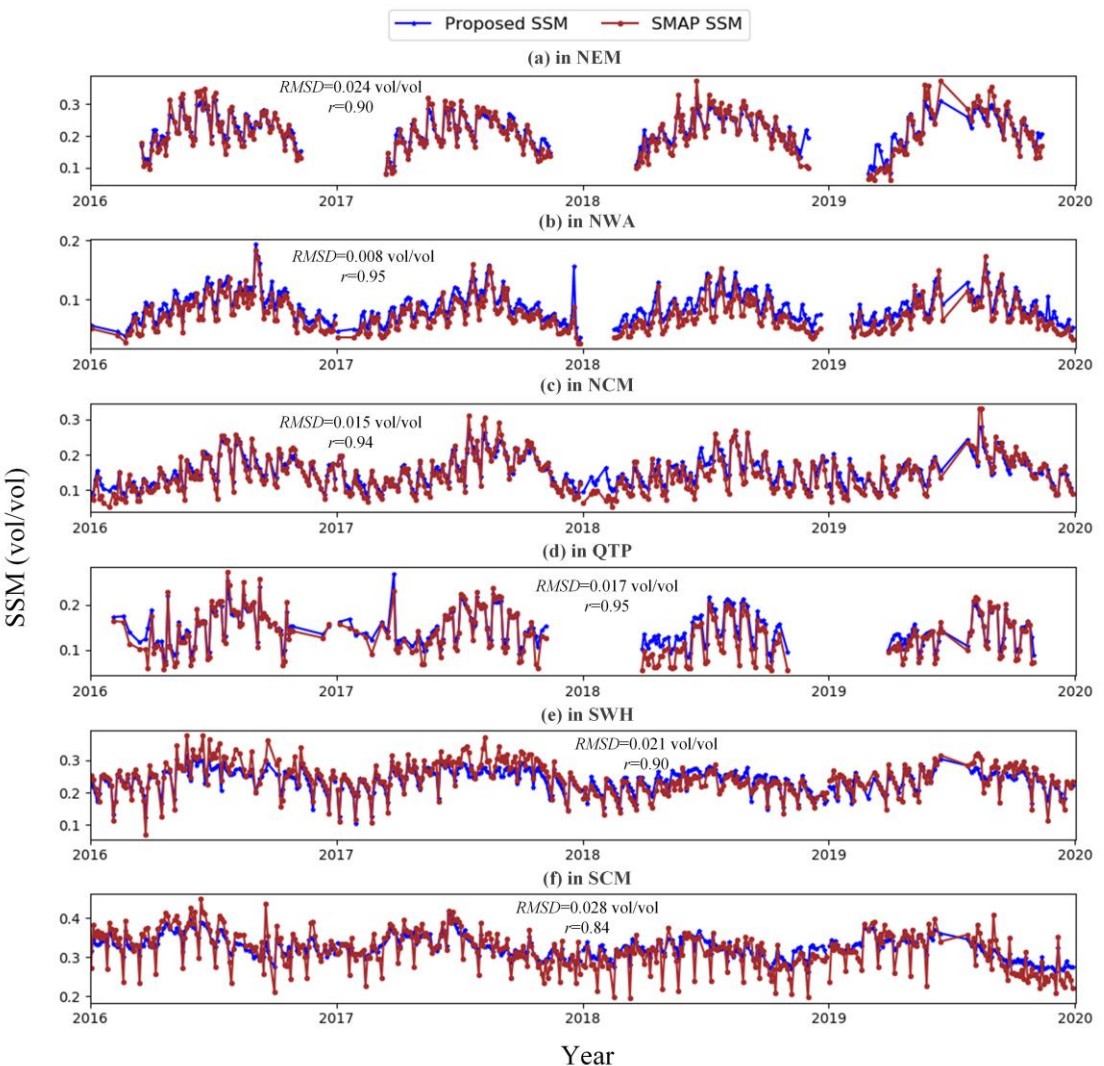

**523**

**524**         Fig. 5 Time series of SSM aggregated at each of the 6 different geographic-climate regions (as

**525** shown in Fig.1) in China for our developed 1-km product as well as for the SMAP 36-km SSM dataset.

**526**             The time series range from 2016 to 2019, with a revisit cycle of three days.

**527**      In Fig. 6-(b) we employed the downscaled SSM image on May 29, 2018, as an

**528** example to demonstrate the spatial features of the developed product. Meanwhile, we

**529** also show the map of SMAP/Sentinel combined SSM (SPL2SMAP_S_V3) obtained

**530** from May 26 to May 31, 2018 in Fig. 6-(a), as a contemporaneous comparison reference.

**531** Clearly, the SPL2SMAP_S_V3 map has a much lower coverage percentage over the

**532** study region compared with the map of the currently developed product on one single

date, even though the former was generated based on multi-date images. Both maps

show similar spatial texture depicting the relatively dry climate in northwestern China

compared with the humid climate in the Middle-lower Yangtze River Plain.

Nevertheless, there also exist cases where the details in texture differ prominently, like

that in the far northeastern end of the country.

For the sake of further analysis on this point, results of the quantitative comparison

as proposed in Section 2.1.3 and Section 2.1.4, is demonstrated in Fig. 6-(c), -(d), -(e),

and -(f). Fig. 6-(c) and -(d) show the RMSD maps of the two respective products against

SMAP radiometer-based SSM estimates at the 36-km pixel scale. For both products it

is manifested that compared with the lower averaged RMSD of 0.04 vol/vol in the

NWA region, the uncertainty can increase (shown in yellow) in the densely vegetated

NEM and the SCM regions, with averaged RMSDs of 0.07-0.08 vol/vol. However, our

developed product has noticeably lower RMSD (0.05 vol/vol) than the

SPL2SMAP_S_V3 data (0.07-0.09 vol/vol) in the SWH and part of the QTP regions.

Considering their relatively higher elevations, it may be roughly drawn that our

downscaled SSM product is more reliable than that downscaled based on active-passive

microwave combined datasets in areas with increased topographic effects. Fig. 6-(e)

shows that the currently developed SSM product obtained a 0.078 vol/vol ubRMSD

and a correlation coefficient of 0.55 against the in-situ soil moisture measurements. It

converges more apparently to the 1:1 line when compared with validation result of the

SPL2SMAP_S_V3 dataset in Fig. 6-(f). As with the area of China, therefore, the

currently developed product is generally superior to the global SMAP/Sentinel
combined SSM in terms of both coverage percentage and estimate accuracy.

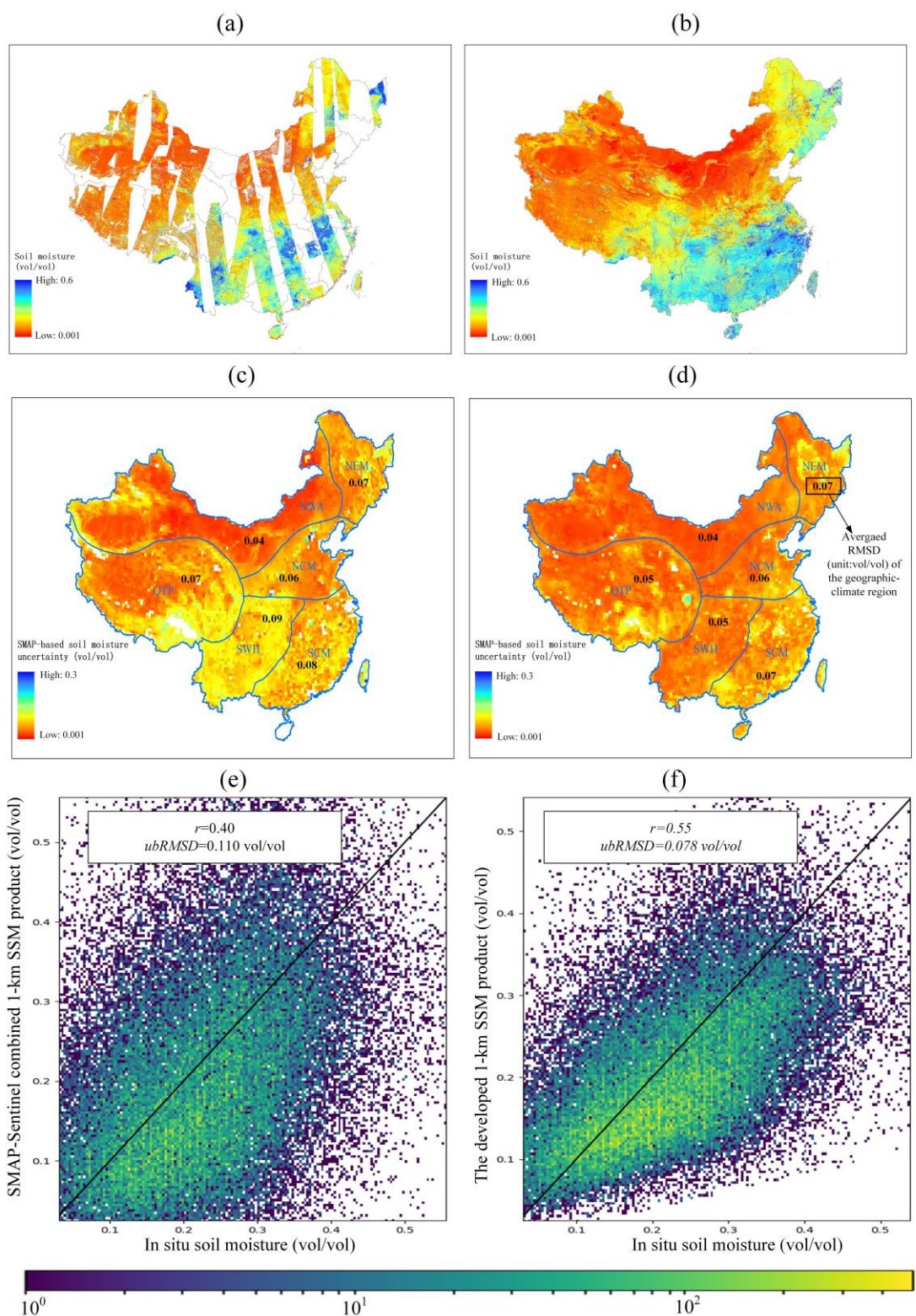

Fig. 6 Comparison results between the currently developed 1-km SSM product and the SMAP/Sentinel
combined 1-km SSM (SPL2SMAP_S_V3). (a) SPL2SMAP_S_V3 SSM images over China at about
6:00 a.m. systhesized by 6 continous dates from May 26, 2018 to May 31, 2018. (b) The SSM image at
1:30 a.m. of May 29, 2018 from the currently developed product. (c) Spatial uncertainty (RMSD) map
of the SPL2SMAP_S_V3 product against SMAP radiometer-based SSM retrievals at the 36-km pixel
scale over China for years of 2017, 2018, and 2019. (d) Same to (c) but for validaiton of the currently
developed SSM product. The black numbers in each of the geographic-climate regions indicate
averaged uncertainty (RMSD, unit: vol/vol) of the region. (e) Validation results of the
SPL2SMAP_S_V3 product against in-situ soil moisture measurements over China for years of 2017,
2018, and 2019. The black solid line is the 1:1 line. (f) Same to (e) but for validaiton of the currently
developed SSM product.
In Fig. 7, we display the cumulative distribution frequency of coverage
percentages of the downscaled SSM product and of the original PM NN-SM product
for each season. We should be noted that in this statistical scheme, pixels identified as
static water body by the MODIS MCD12Q1 land cover type product were not
considered in the denominator of the coverage percentage. Besides, the gap time
between the respective on-orbit period of AMSR-E and of AMSR-2 (from October
2011 to June 2012, during which there are no effective observations from the PM NN-
SM product) were also excluded.  It is apparent that in Fig. 7-(b) and -(c), almost all
downscaled daily SSM images over the 16-17 years have achieved a coverage
percentage higher than 85%. In comparison, the majority of the PM NN-SM daily
images have their coverage percentages below 80% over the study region, primarily
due to the PM-seam gaps particularly existing in low latitudes (see Section 2.2.2). In
Fig. 7-(a) and -(d), the percentages of effective pixels in both the PM and the
downscaled SSM images are far lower than their counterparts in the other two

**582** subfigures. This is mainly ascribed to extreme meteorological conditions including

**583** snow, ice, and frozen soils that are typically persistent throughout most of these

**584** specified months in the northwestern regions of China. Such conditions can impede

**585** reliable estimates of SSM based on all satellite remote sensing techniques in the current

**586** time. The above inter-seasonal differences on data coverage are also reflected in Fig. 8

**587** in another manner based on presenting the spatial distributions of number percentages

**588** of available dates in each three-month period.

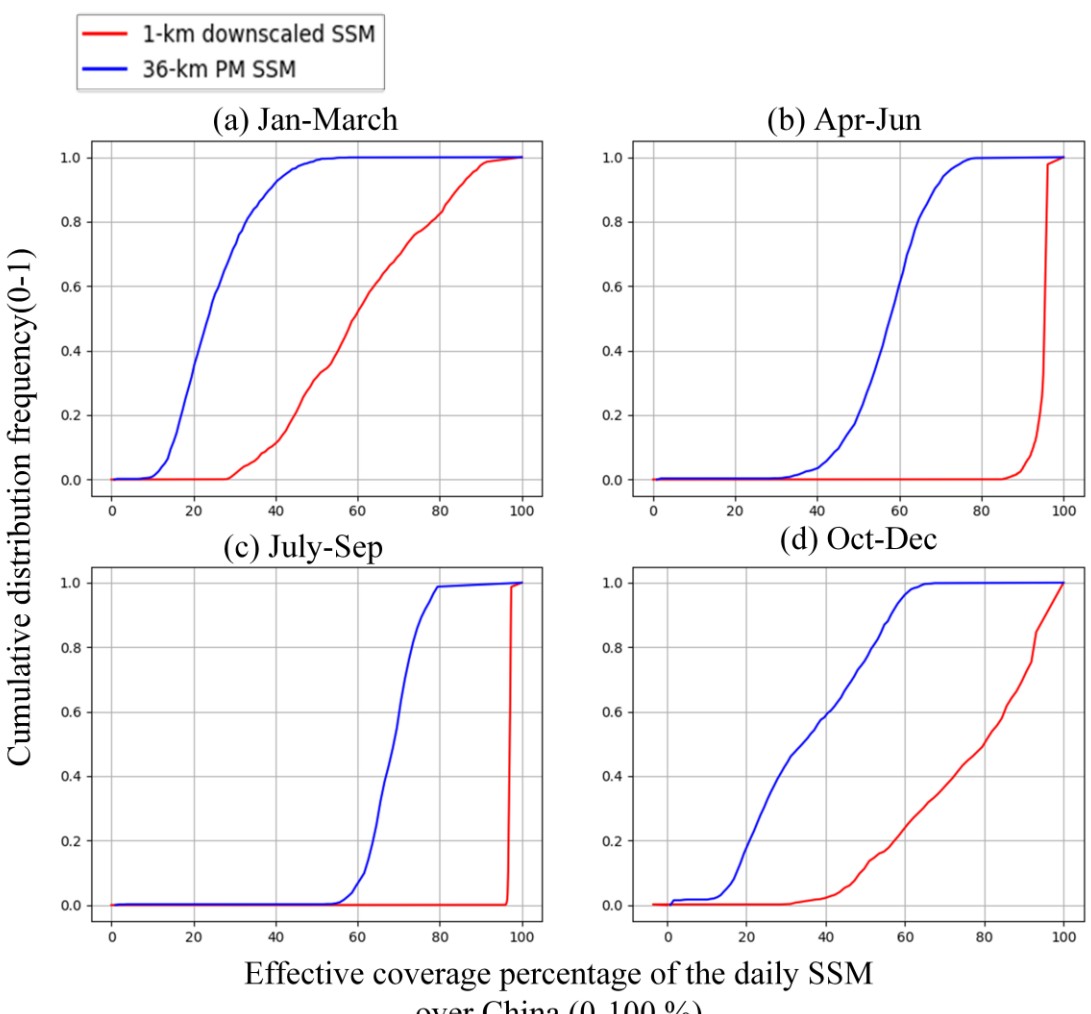

**589**

**590**    Fig. 7 Cumulative distribution frequency of our proposed SSM product against the original 36-km SSM

**591**    product for different seasons. The period between October 2011 and June 2012 is excluded in the

**592**    current statistics.

**593**

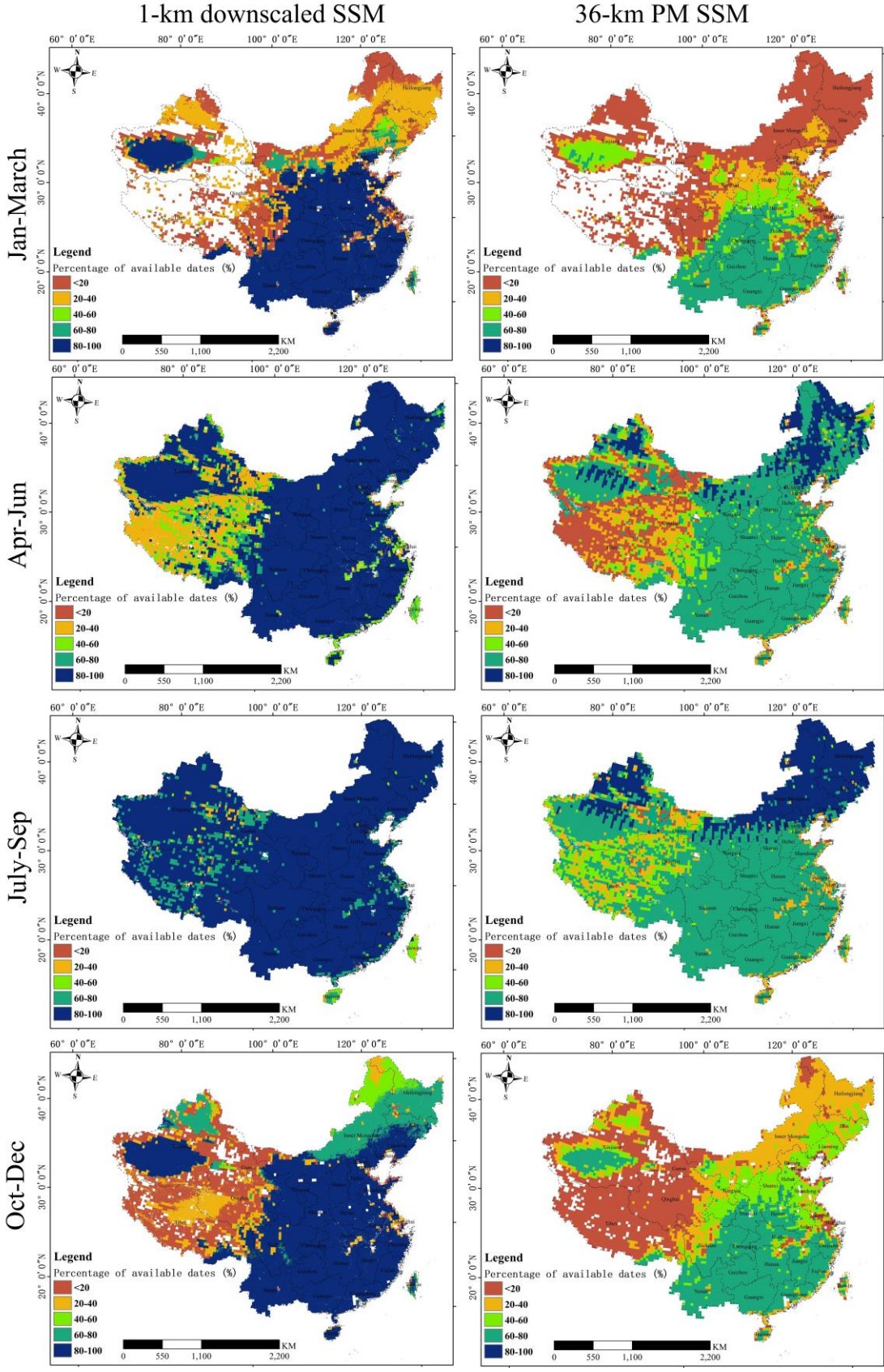

Fig. 8 Spatial distributions on percentage of day numbers with available estimates for the currently

developed 1-km SSM product and the original 36-km PM data during 2003-2019. The four different

**597** periods (i.e., January-March, April-June, July-September, October-December) of a year are treated

**598** respectively. The period between October 2011 and June 2012 is excluded.

**599** The techniques behind coverage improvement of the downscaled SSM (against

**600** PM and optical data inputs) can be categorized into two classes, i.e. cloud gap-filling

**601** of the input optical datasets (see Section 2.2.1), as well as the filling of downscaled

**602** SSM in PM-seam gaps (see Section 2.2.2). Table 2 reports the specific validation results

**603** (using averages of ground measurements at all stations) of downscaled SSM in these

**604** coverage-improved conditions, relative to that generated without using any coverage

**605** improvement technique, in order to evaluate the propagated effect of such techniques

**606** on the final product. The very limited difference for ubRMSD values (0.053 vol/vol

**607** versus 0.056 vol/vol) between cloudy and clear-sky conditions suggest that the 1-km

**608** SSM estimates from our final product are generally compatible between cloudy and

**609** clear-sky conditions. The downscaled SSM estimated for regions of PM-seam gaps

**610** have a slightly worse (but still acceptable) accuracy, considering its ubRMSD of 0.059

**611** vol/vol compared to the 0.052 vol/vol ubRMSD of the PM-observed 1-km pixels. In

**612** summary of Fig. 7 and Table 2, the currently developed product has achieved a

**613** substantially improved spatial coverage against the original remote sensing input

**614** datasets, whilst successfully preserved the SSM downscaling accuracy of the

**615** observation-covered pixels at the same time.

Table 2 Comparisons between validation results for pixels under coverage-improved regions and

for pixels under remote-sensing-observation-covered regions.

| Evaluation metric[*] | Comparison between cloudy and clear-sky conditions | | Comparison between passive microwave (PM) observed regions and regions of PM-seam gaps | |
|---|---|---|---|---|
| | Clear-sky condition | Cloudy condition | PM-observed regions | PM-seam gaps |
| ubRMSD (vol/vol) | 0.053 | 0.056 | 0.052 | 0.059 |
| Correlation coefficient | 0.49 | 0.47 | 0.49 | 0.44 |

[*]All evaluation metrics in this column indicate the average of all available stations

## 4. Discussion

## 4.1 Uncertainty on SSM evaluation between satellite- and ground- scales

In this study, we made evaluations on remote sensing SSM products at different spatial resolutions, using measurements from 2000+ stations provided by the national-level soil moisture observation network of China as standard benchmark. Through the evaluations, a ubRMSD of 0.074 vol/vol is reported for the original 36-km NN-SM SSM product (Fig.A1-b). We notice that this result is considerably poorer if compared with another previous evaluation campaign targeting at the same product (Yao et al., 2021), which achieved a global RMSE (RMSD) of 0.029 vol/vol. However, this difference is not unexpected because the two campaigns were carried out in different regions of the world. Also, that particular study (Yao et al., 2021) was conducted based on completely different ground soil moisture observations provided by the International

Soil Moisture Network (ISMN) (Dorigo et al., 2021). Compared to the observation
network employed in this study, the observation sites of ISMN are more intensively
distributed as an "integrated soil moisture station" so as to provide spatially average
soil moisture within a grid of tens of kilometers. In this regard, we admit that the ISMN
is generally more professional in evaluating satellite PM-based SSM retrievals at a
coarser resolution. But on the other hand, only a few ($\leqslant$4) of such "integrated stations"
have been set up sporadically within China, making the ISMN data much less
representative of our study region compared with the national-level soil moisture
network of China exploited by our current study.
Although the higher RMSD of the national-level soil moisture network of China
may indicate larger measurement uncertainty than the ISMN, the negative influence
that might be imposed on our study purpose should be inconsequential. This is because
we focus more on the relative validation performance of different SSM products, rather
than on the absolute value of any evaluation metric including ubRMSD and correlation
coefficient calculated against ground measurements. Specifically, the 1-km downscaled
SSM obtained an average ubRMSD of about 0.054 vol/vol among different stations
according to Fig. 4-(b). Besides, result of the evaluation in Fig. 6-(d) based on
combination of multi-station ground measurements shows a global ubRMSD of 0.078
vol/vol for this product. Overall, the above-mentioned results can be identified as at
least comparable to the global (multi-station based) ubRMSD of 0.074 vol/vol of the
original NN-SM data as they are evaluated against the same benchmark. Therefore,
conclusion is safely drawn that the currently developed product preserves the retrieval

**654** accuracy of the coarse-resolution NN-SM data, whilst improving the spatial

**655** representativeness of the latter product substantially according to the mostly positive

**656** $G_{down}$ values in Fig. 4-(a).

**657**     Moreover, one may also argue that the $r$-value of 0.55 for the currently developed

**658** product in Fig. 6-(d) is not sufficiently high compared with several previous studies

**659** (Wei et al., 2019; Sabaghy et al., 2020) obtaining $r$-values above 0.7 for temporal

**660** analysis of satellite remote sensing soil moisture. However, we should be noticed that

**661** these previous studies have conducted analyses respectively at the temporal and the

**662** spatial dimensions. Based on their results, the spatial analysis typically derived lower

**663** $r$-values ($<$0.4) compared to that at the temporal dimension. This is probably because

**664** the heterogeneity degree of remote sensing pixels can vary significantly across different

**665** sites. Since the evaluation in Fig. 5-(d) was deployed at the 'spatio-temporal' integrated

**666** dimensions, such an $r$-value is expected. This is also close to the global $r$-value of 0.6

**667** for validation of the coarse-resolution NN-SM product as reported in Yao et al. (2021).

**668** ## 4.2  Uncertainty on cloud gap-filling and validations of LST

**669**     As mentioned in Section 2.2.1, LST gap-filled based on the STDF method was

**670** used alone as one of the main input datasets for SSM downscaling under cloudy weather.

**671** Although such LST inputs contain clear-sky bias from the real cloudy condition, it

**672** performs better in driving the SSM downscaling model compared with its bias-adjusted

**673** counterpart (see Section Appendix-B for details). The reason may be linked to one of

**674** the basic theories behind our SSM downscaling methodology, i.e. the "universal

**675** triangle feature space (UTFS)" theory (Carlson et al., 1994). In the UTFS, clear-sky

LST is employed to implicitly quantify the surface soil wetness degree as it correlates
with the dynamics of soil evaporative efficiency and soil thermal inertia when
vegetation cover density is fixed. Under cloudy conditions, however, the satellite
observed LST is subjected to not only surface soil property, but also to that related to
cloud insulation effect from solar incoming radiation and ground long wave outgoing
radiation. As a result, the actual relationship between SSM and cloudy LST could be
much more complicated than the one that has been described by the UTFS-based SSM
downscaling model (i.e. Equation-2). In comparison, LST generated by the STDF alone
for assumed clear-sky conditions, as is free from interference of cloud, would be a
comparatively more competent input variable for driving the UTFS-based SSM
downscaling model under non-rainy clouds. This is especially the case for thin and
short-time clouds with marginal direct feedbacks on surface soil wetness.
However, we admit that the STDF-filled LST under rainy clouds is also not suitable
for our study purpose. This may explain the slightly higher RMSD for SSM under cloud
based on STDF-filled LST (0.056 vol/vol) compared to that under real clear sky (0.053
vol/vol), as shown in Table 2. In reality, the actual negative influence of cloud on the
final SSM product may be even more serious than indication from the above RMSD
difference (i.e. 0.056-0.053 = 0.003 vol/vol), due to the portion of "clear/cloudy-
weather-mixed" spatial windows during the fitting process of the downscaling model.
In these windows, uncertainty in cloud gap-filled LST may affect accuracy of the fitted
model coefficients and thus deteriorate the final SSM estimates in clear-sky pixels
within the same window. Consequently, the above RMSD difference has been more or

**698**    less underestimated. Despite all of above, in our study area of China we regard the

**699**    STDF-filled LST as a more optimal proxy of heat flux for estimating SSM under clouds,

**700**    compared to the bias-adjusted LST. On the other hand, future efforts are encouraged to

**701**    further clarify the mechanical relationships between STDF-filled/bias-adjusted LST

**702**    and soil wetness degree under clouds.

**703**        Different from a number of previous studies (Jiménez et al., 2017; Dowling et al.,

**704**    2021; Yang et al., 2019) validating satellite thermal-infrared-based LST based on

**705**    longwave radiation observations made at footprint-level observation stations (e.g. flux

**706**    towers), our study has used 0-cm top ground temperatures as the primary benchmark

**707**    for this validation campaign instead. Similar to that for SSM validation, the most crucial

**708**    motivation driving such an experimental design is the significantly intensive

**709**    distribution of the meteorological stations compared to the very limited number of

**710**    active and effective flux towers available in China. It is noted that these measurement

**711**    devices at all of the meteorological stations are required to have been instrumented

**712**    under open environmental conditions with relatively lower fraction of tall trees and

**713**    water bodies, in order to conduct efficient monitoring at the physics of near-surface air.

**714**    This can also be reflected in Fig.4-(c), which reveals no stations built within forest

**715**    covers. Moreover, as we only focus on the mid-night scenario when the states of all

**716**    land observations are "most stable" during one diurnal cycle, uncertainties due to the

**717**    possible temperature inconsistency between bare ground surface and high tree surface

**718**    as well as due to the temporal mismatch (from about 1:30 to 2:00 A.M.) should have

**719** marginal effect on our results. We have carried an extra test that can confirm this

**720** discussion, with the detailed procedures described in Section Appendix-C.

## 4.3 Major novelty, unique profit, and future prospect of the developed product

**721**

**722**

**723** Compared with the widely known active/passive microwave combined SSM

**724** product (e.g. the SPL2SMAP_S_V3) and other PM/optical-data combined counterparts

**725** which were also published recently but at the monthly scale (Meng et al., 2021), the

**726** major novelty of the currently developed product mainly lies in the fact that it has

**727** achieved progress on all of the three crucial dimensions of satellite remote sensing,

**728** including the temporal revisit cycle (daily), the spatial resolution (1-km), and the quasi-

**729** complete coverage under all-weather conditions. To our knowledge, this has rarely been

**730** achieved by previously developed satellite soil moisture product at regional scales. For

**731** realization of the above-mentioned progresses, we have fused the SSM downscaling

**732** framework with other techniques including cloud gap-filling of thermal infrared LST,

**733** MWS-based temporal filtering of vegetation indices, as well as reconstruction of seams

**734** between neighboring PM swaths in low latitudes. The final SSM estimates under cloudy

**735** conditions and intersected with the PM-seam gaps were specially validated against the

**736** rest estimates under clear sky and in the regions covered by PM observations,

**737** respectively (Table 2). The comparable performances among all treatment groups

**738** herein confirm that the accuracy of the product is stable and consistent among all

**739** weather conditions.

**740**       With improvement achieved at the three dimensions, unique profit of the currently

**741** developed product can be taken by subsequent studies and various industrial

**742** applications. For example, the capability of this product can be investigated on

**743** capturing the short-term anomaly of local hydrological signals as well as improved

**744** monitoring on drought disasters, which used to be investigated mainly at a coarser

**745** resolution by PM SSM (Scaini et al., 2015; Champagne et al., 2011; Albergel et al.,

**746** 2012). For another, taking advantage of its all-weather daily time series, the product

**747** can be utilized together with precipitation data to isolate and quantify the anthropic

**748** influence on regional water resources from the natural hydrological dynamics.

**749** Examples of such anthropic signals include agricultural irrigation activities, as well as

**750** finer-scale information on agricultural crops which was previously interpreted based on

**751** PM-driven techniques (Song et al., 2018). In addition, we should realize the important

**752** role of soil moisture as a constraint for accurate estimation of surface

**753** evapotranspiration and runoff (Zhang et al., 2020; Zhang et al., 2019). Therefore, the

**754** profit of this product can be further enhanced if coupled with land-atmosphere coupled

**755** models to produce new insights into water-cycle processes of earth surface at a finer

**756** spatio-temporal scale.

**757**       There are still some limitations on our current product to be further improved. First,

**758** there may exist the 'mosaic effect' at the original PM (36-km) pixel edge. As mentioned

**759** in Section 2.2.2, we have used a parameter of 'spatial square window (*ws*)' in Equation-

**760** (3) to minimize this negative effect. However, it is still difficult to utterly avoid such

**761** negative effect. This is a challenge for all existing SSM downscaling methods (Molero

**762** et al., 2016; Stefan et al., 2020; Peng et al., 2016), especially considering the large

**763** spatial scale of our study and all uncertainties discussed in Sections 4.1 and 4.2. Besides,

**764** other negative influences can be imposed by potential imperfections identified from the

**765** original PM product, e.g. from PM SSM retrievals in the QTP region with complicated

**766** topography, melt snow or partially frozen soils that cannot been completely screened

**767** out by the original product flag in winter. For these extreme conditions, the accuracy

**768** of the downscaled SSM may need further validation campaigns like field surveys and

**769** experiments, based on which the data quality flag can be better built for the product's

**770** futural version.

**771** The methodological framework proposed in this paper is prospective to be

**772** universally applied in other regions of the world to serve for better monitoring of the

**773** global surface wetness in the following studies. If applied in continental and global

**774** scales, however, the current process for gap-filling of PM seams may require further

**775** attention and improvement. In this study, SSM in regions intersected with PM-seam

**776** gaps were estimated using TB observations from PM swaths at neighboring dates (see

**777** Equation-5). Although the errors in the PM-seam gaps over China as reported by Table

**778** 2 are only slightly larger compared to the PM-covered regions, they cannot be ignorable

**779** completely and may leave extra concern on the universality of this technique, especially

**780** in the low latitudinal tropical regions where the effect of PM-seam gap is more apparent

**781** than in our study area. Besides, another imperfection of this data product lies in the gap

**782** period between AMSR-E and AMSR-2. Considering the different systematic error

**783** patterns of various PM SSM products, we did not generate downscaled SSM based on

**784** other PM products (e.g. the SMOS SSM product) during this period but just left the

**785** period as null values. We suggest a more rigorous and universal inter-calibration

**786** framework on different PM SSM products to be developed in the future for a long-term

**787** consistent 1-km downscaled SSM dataset.

## 5. Conclusions

**789** This paper describes the main technical procedures of a recently developed remote

**790** sensing surface soil moisture (SSM) product over China covering the recent ten years

**791** and more. Based on combination of passive microwave SSM downscaling theory and

**792** other related remote sensing techniques, the product achieves multi-dimensional

**793** distinctive features including 1-km resolution, daily revisit cycle, and quasi-complete

**794** all-weather coverage. These were rarely satisfied completely by other existing remote

**795** sensing SSM product at regional scales. Validations were conducted against

**796** measurements from 2000+ automatic soil moisture observation stations over China.

**797** Overall, an average ubRMSD of 0.054 vol/vol across different stations is reported for

**798** the currently developed product. The mostly positive $G_{down}$ values show this product

**799** has significantly improved spatial representativeness against the 36-km PM SSM data

**800** (a major source for downscaling). Meanwhile, it generally preserves the retrieval

**801** accuracy of the 36-km data product. Moreover, additional validation results show that

**802** the currently developed product surpasses the widely used SMAP-sentinel combined

**803** global 1-km SSM product, with a correlation coefficient of 0.55 achieved against that

**804** of 0.40 for the latter product. At the regional scale, time series patterns of our developed

**805**     data product are highly correlated with that of the widely recognized SMAP radiometer-

**806**     based SSM for all geographic settings. The methodological framework for product

**807**     generation is promising to be applied at the continental and global scales in the future,

**808**     and the product is potential to benefit various research/industrial fields related to

**809**     hydrological processes and water resource management.

**810**

## Appendix

## A. Evaluation on different PM SSM products

We have made evaluations on the various AMSR-based SSM products (as shown in Table 1) covering the recent 10 years or longer, based on our soil moisture observation network all over China. The SMAP radiometer-based SSM dataset, as described in Section 2.1.4, was also evaluated as a reference. The evaluation period covers the three years of 2017, 2018, and 2019. All AMSR-based 25-km grids were re-set to the SMAP 36-km grid system using the nearest resampling method. Only grids that contain equal or more than 4 soil moisture measurement stations were employed, in which, the grid-based PM SSM estimate was compared with average of measurements from all interior stations. Finally, 53 grids were selected, as shown by the green color in Fig.A1-(g). For AMSR-based products, only the mid-night descending datasets were evaluated, whist for the SMAP product, our evaluation only focused on its descending mode in the early morning.

As manifested by Fig.A1-(a) to -(f), the selected SSM product in the current study, i.e., the NN-SM product has an unbiased RMSD of 0.074 vol/vol and a correlation coefficient of 0.49. This obviously outperforms the other three traditional AMSR-based SSM products (i.e. JAXA-AMSR, LPRM-AMSR, and UMT-AMSR products) and is only inferior to the SMAP SSM retrievals, whilst the later only covers the latest period since 2015. As far as CCI data are concerned, it has a similar performance against the selected NN-SM in general. Nevertheless, the region marked by red circle in Fig.A1-

(c) indicates that CCI estimates have a considerably larger proportion of overestimated
anomalies. But overall, the primary reason that we have abandoned CCI but selected
NN-SM is because the latter can provide a higher coverage fraction of valid pixels in
our study region, as has been stated in Section 2.1.1.

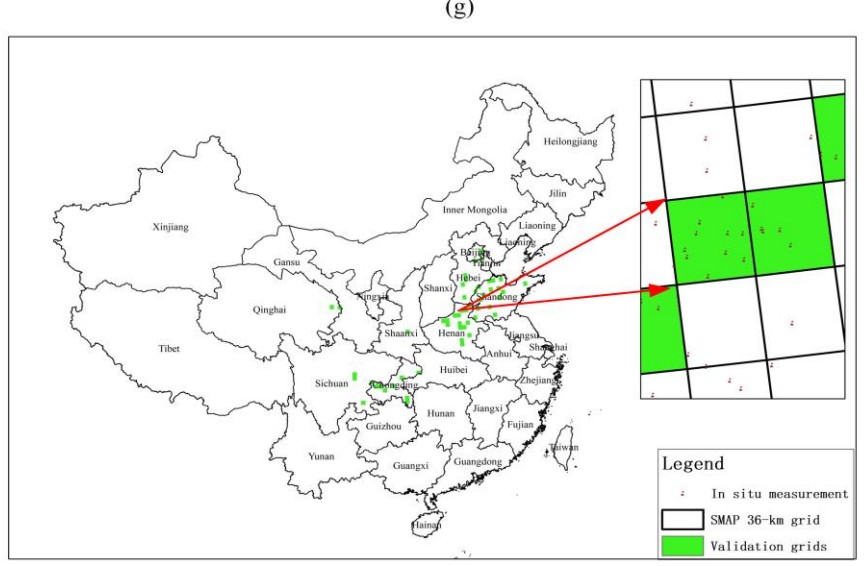

(a) SMAP

*ubRMSD*=0.065 vol/vol, *r*=0.60

(b) NN-SM

*ubRMSD*=0.074 vol/vol, *r*=0.49

(c) CCI

*ubRMSD*=0.075 vol/vol, *r*=0.48

(d) JAXA-AMSR

*ubRMSD*=0.095 vol/vol, *r*=0.15

(e) LPRM-AMSR

*ubRMSD*=0.077 vol/vol, *r*=0.42

(f) UMT-AMSR

*ubRMSD*=0.095 vol/vol, *r*=0.19

Passive microwave remote sensing soil moisture (vol/vol)

In situ soil moisture (vol/vol)

(g)

Legend

In situ measurement

SMAP 36-km grid

Validation grids

Fig. A1 (a)-(f) Comparison of different PM SSM products (as reported in Table 1) against the in situ
SSM measurements in China. (g) Locations of the 36-km EASE-GRID-projection based pixels used for

this comparison campaign.

## B. Evaluation on the influence of bias adjustment for reconstructed 'clear-sky' LST under cloud

In Section 2.2.2, we have emphasized that the gap-filled LST for cloudy pixels
reflects the theoretical surface temperature of that pixel under a hypothetical clear-sky
condition. As this cloud gap-filled LST would suffer from a possible bias against the
real surface temperature under cloud (Dowling et al., 2021), we made an additional
experiment regarding to further improvement of this cloud gap-filled LST. The follow-
up step for bias adjustment of this hypothetical clear-sky LST (but actually under
cloudy conditions), as expounded in Section 4.2 of Dowling et al. (2021), was
conducted herein using remote sensing and in situ LST data over China but only in
2018. We illustrate the validation results for bias adjusted and non-bias adjusted LST
under cloudy conditions in Fig. A2-(b) and -(c), respectively. Similar to Fig. 3,
validation results for clear-sky LST of that year are also displayed (Fig. A2-(a)) for
comparison. The results generally show that the follow-up step is effective in reducing
the bias of the originally gap-filled 'clear-sky LST' under cloudy conditions (from -1.7
K to 0.4 K).
In the subsequent step, we substituted the original non-bias adjusted LST under
cloudy conditions with its bias adjusted counterpart, and used the latter as the input for
SSM downscaling. The general validation results of the downscaled SSM are illustrated

**859** in Fig. A3 (similar to that presented in Fig. 4-a and -b). Contrary to the above-analyzed

**860** Fig. A2, the bias adjusted cloudy LST with better gap-filling accuracies, however,

**861** obtained inferior performance in SSM downscaling. This final validation result, to

**862** some degree, confirms our assumption in Section 2.2.2 that the reconstructed cloudy

**863** LST but for the hypothesized clear-sky condition is the better proxy of surface moisture

**864** dynamics. But overall, as all LST estimates discussed herein are for the midnight

**865** scenario (when the energy interaction between atmosphere and land surface is relatively

**866** weak), the RMSD difference for different weather conditions in Fig.A2 is expectedly

**867** marginal. As a consequence, the difference in ubRMSD of SSM in Fig.A3 can hardly

**868** be identified as 'very significant'. Therefore, we encourage further tests on this

**869** conclusion in specific future studies to confirm its universality, especially for situation

**870** of the 'morning to noon' time window.

**871**

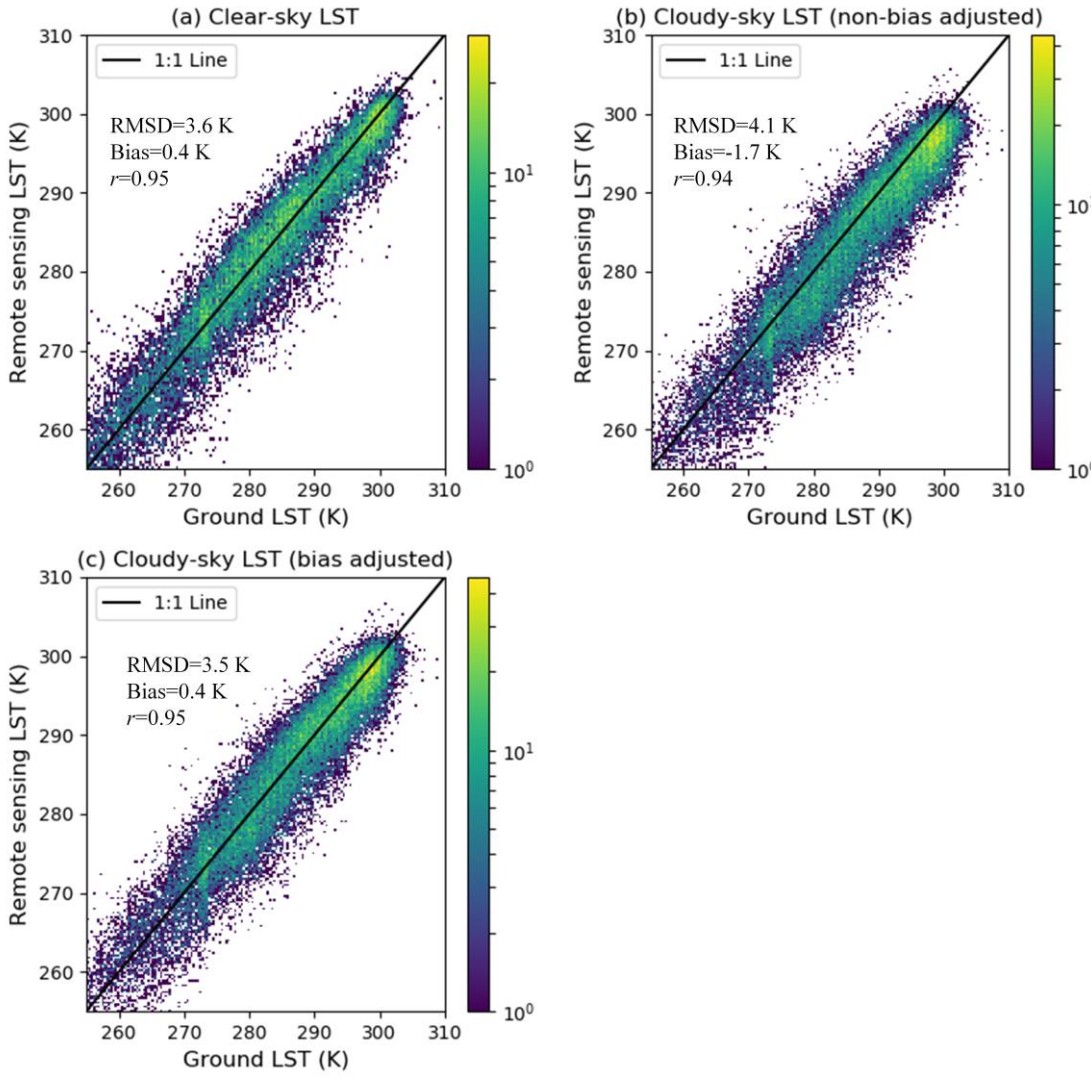


Fig. A2 Validation of the clear sky LST (a), reconstructed LST under cloud but with no passive-
microwave based bias adjustment (b), as well as the reconstructed LST under cloud with passive-
microwave based bias adjustment (c) respectively, based on the 0-cm ground temperature

measurements at meteorological stations.


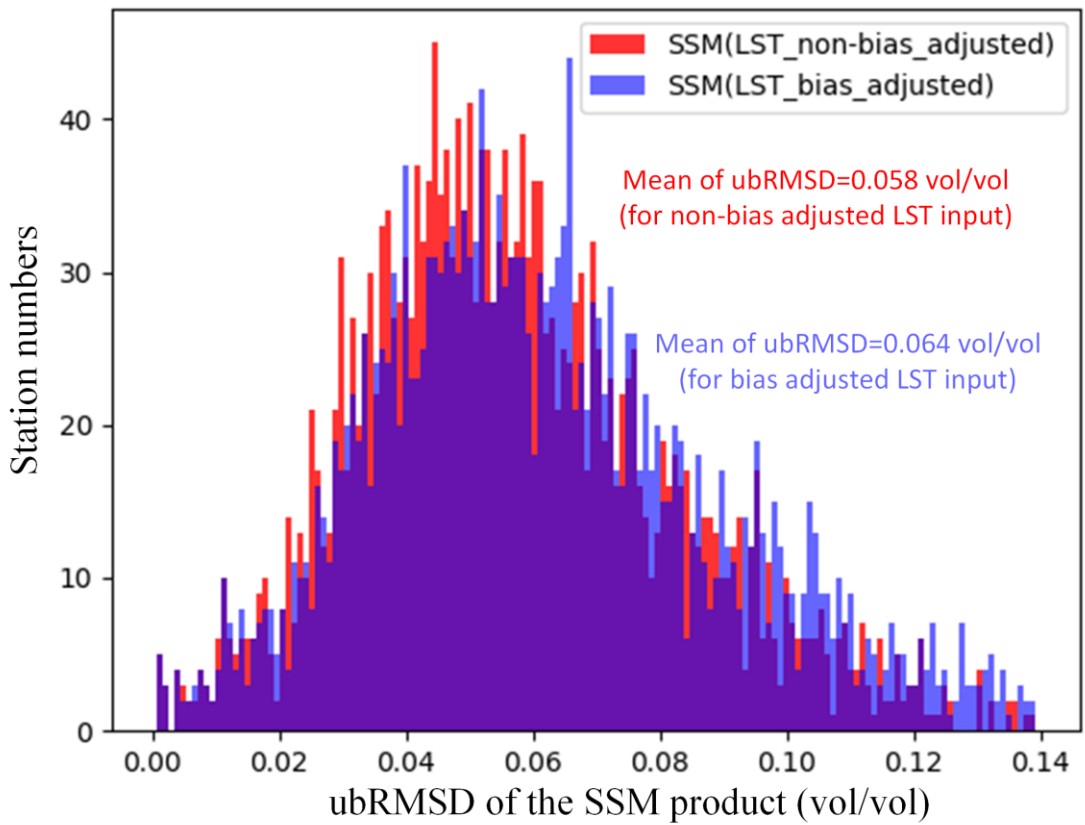

Fig. A3 The statistical distribution of ubRMSD at different stations for SSM estimates driven by two

respective kinds of cloudy LST inputs.

## 881 C. Uncertainty test between 0-cm ground temperature
## 882 observations and flux-tower-derived thermal infrared LST

We herein utilized 4 flux towers to calculate their footprint-level (about 500-1000
m) thermal infrared LST based on long wave radiation measurements, plus broad band
emissivity data derived from the MODIS MYD21A1 product (MYD21A1N.V061).
The 4 towers are all characterized by moderate or low vegetation (grassland) and are
dispersedly located at different eco-regions of China, namely the towers of Changling,
Huailai, Yakou, and Naqu (see the inset map in Fig.A4-b). Data from Changling are
derived from the FLUXNET community (FLUXNET2015 Dataset - FLUXNET ) in 2010.
Data from the other three towers are derived from the National Tibetan Plateau Data
Center, with data DOIs of http://dx.doi.org/10.11888/Meteoro.tpdc.271094 for Huailai
in 2018, http://dx.doi.org/10.11888/Meteoro.tpdc.270781 for Yakou in 2018, and
http://dx.doi.org/10.11888/Meteoro.tpdc.270910 for Naqu in 2016. These data have
been preprocessed by their providers to record the dynamics of those variables at a half-
hour interval. The algorithm for calculating LST based on flux-tower-derived long
wave radiation is inherited from Wang and Liang (2009). We first compared the flux-
tower-derived night-time LST estimates between 1:00-1:30 A.M. and 1:30-2:00 A.M..
As shown by Fig.A4-(a), the very slight RMSD of 0.72 K suggests that LST is generally
stable between 1:00 and 2:00 A.M. at night. In Fig.A4-(b), we also found marginal bias
and RMSD within 1 K between average flux-tower-derived LST of 1:00- 2:00 A.M.
and the corresponding 0-cm ground temperature at close meteorological sites (within 1
km and at 2:00 A.M.).
In Fig.A4-(c) we demonstrate time series for monthly average NDVI (derived as
in Section 2.2.1) at the 1-km pixels containing each of the four sites from 2003-2019.
Clearly, there are very rare cases with NDVI values exceeding 0.5, corroborating the
"open environmental conditions" met by the meteorological stations. In view of above,
it is feasible for our study to have used the 0-cm ground temperature at pixels of such
moderate to low vegetation covers as the evaluation benchmark of the satellite-derived
thermal infrared LST.

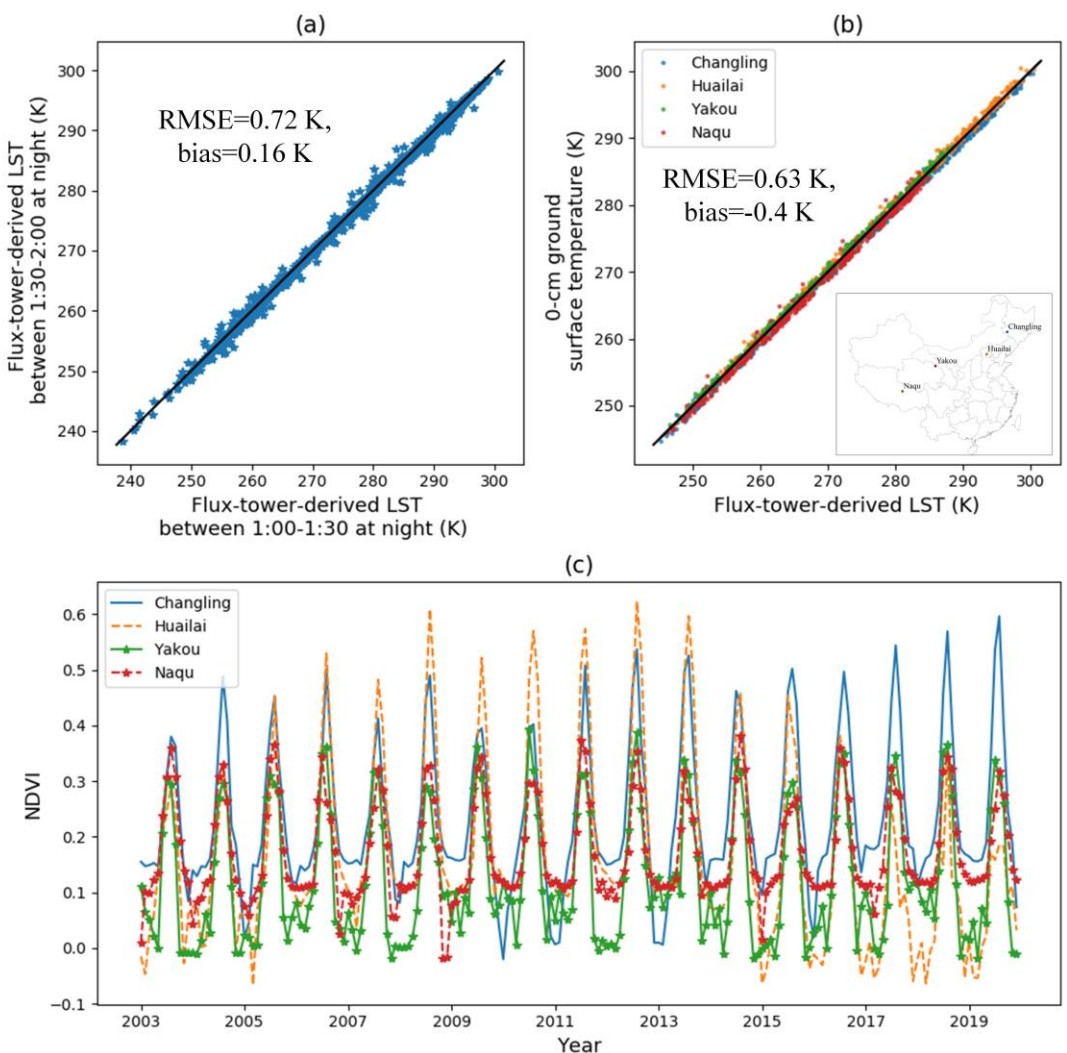


Fig. A4 (a) Comparison of LST between 1:00-1:30 A.M. and 1:30-2:00 A.M. for the four selected flux
towers. (b) Comparison of flux-tower-derived LST averaged for 1:00-2:00 A.M. at the four towers and
corresponding night-time 0-cm ground temperature at proximal meteorological stations. The inset map
shows the location of the four flux towers. (3) Monthly NDVI time series for 1-km pixels containing
each of the four flux towers.

**Author contributions**
Peilin Song and Yongqiang Zhang designed the research and developed the whole
methodological framework; Peilin Song and Yongqiang Zhang supervised the
processing line of the 1-km SSM product; Jianping Guo and Bingtong provided in situ

soil moisture data for validation; Peilin Song wrote the original draft of the manuscript;

Yongqiang Zhang, Peilin Song, Jiancheng Shi, and Tianjie Zhao revised the manuscript.

## Competing interests

The authors declare that they have no conflict of interest.

## Data availability

The published SSM dataset is available under the Creative Commons Attribution

4.0 International License at the following link:

http://dx.doi.org/10.11888/Hydro.tpdc.271762 (Song and Zhang, 2021b). This dataset

covers all of China's terrestrial area at a daily revisit frequency (about 1:30 A.M. at

local time) and a 1km spatial resolution from January 2003 to October 2011 and from

July 2012 to December 2019.

## Acknowledgement

The authors would like to thank the National Aeronautics and Space

Administration (NASA) for providing all MODIS and DEM data sets free of charge.

## Financial support

This study was jointly supported by the National Natural Science Foundation of

China (Grant No. 42001304), the Open Fund of State Key Laboratory of Remote

Sensing Science (Grant No. OFSLRSS202117), CAS Pioneer Talents Program, CAS-

CSIRO International Cooperation Program, and the International Partnership Program

of Chinese Academy of Sciences (Grant No. 183311KYSB20200015).

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
