# Peer review of "A 1-km daily surface soil moisture dataset of enhanced coverage"

_Earth System Science Data, 2021_

## Referee Comment (RC1)

**General Comments**

This paper proposes fine-resolution surface soil moisture (SSM) data over China. The significance and potential impact are clear, and the novelty and results are promising. However, a major revision is needed to address my concerns. Please see the comments below.

**Specific Comments**

1) Line 236: About the LST validation:

Aqua nighttime passing time can be 1-hour away from 1:30 am LT. Even nighttime LST does have small variations than daytime LST, can you find some sites with minute-level observations in China to prove that using ground observations at 2:00 am introduces little uncertainty to the validation results?

Besides, the 0-cm ground temperature is different from LST physically, especially over vegetated areas, where SSM estimation by LST may have considerable uncertainty. Over these places, LST is closer to vegetation canopy temperature (air temperature).

To address my two concerns above, I would recommend including a brief test in the discussion by using site-measured LSTs that are computed by surface upward longwave radiation and BBE, and it would be convincing to include sites over various land cover types.

2) is there any evidence to prove the rationality of '7x7' and '-5$^{th}$ to 5$^{th}$'?

3) Fig 3: Clear bias is still shown in filled LST results (Fig 3b) compared to the clear-sky validation (Fig 3a). Will it affect the SSM estimation when clear-sky (unbiased) and filled LSTs (biased) simultaneously exist in a spatial window using the 'universal triangle feature' or in SEE calculation?

4) Fig 5: After readers notice the clear differences between two data at some locations (Fig 5a&b), they may want to know which data is more accurate.

In order to address this concern, you may need to focus on the sites over these regions, where the proposed data have considerable differences with SMAP-Sentinel (e.g. far northeastern, northern west, southern provinces near the sea), specifically and separately, rather than just over entire China (Line 499).

Besides, SMAP shows a very good accuracy (Fig A1a) while the downscaled SMAP-Sentinel (Fig 5c) has large (nearly doubled) ubRMSD. Can you explain why the accuracy is considerably decreased after downscaling?

5) Appendix B:

It's strange that filled LST with considerable bias (-1.7 K) can achieve better SSM accuracy (0.058 vol/vol) than the SSM (0.064 vol/vol) from more accurate/realistic cloudy-sky LST in Fig. A3, and such accuracy difference is even larger than its difference with the clear-sky SSM (0.053 vol/vol, LST is unbiased). If that is the case, the logic behind it is that SSM is not sensitive to the LST, which is not right.

Besides, the LST bias explanation in Lines 309-311 is not convincing: if the filled LST has clear bias compared to site observations, it only means it cannot reflect the realistic surface condition.

**Technical Corrections**

Lines 50-56: references are necessary for the background knowledge introduction, especially for the potential application examples

Line 87: "universal triangle feature (UTF)" or "triangle feature space (TFS)"?

Line 91: please define the acronym UCLA

Line 109: ',' should be removed

Lines 78-80, 112-113: references, please.

Line 117: 'whilst .. even inferior' is not appropriate here. There is no such logic in the context unless you mean 'UTF-based methods are found even inferior to the DISPATCH in a typical humid region'

Lines 124: the objectives you mentioned here are more like broad impacts or potential significance while the objective of a study should be specific.

Line 144: 'after' or 'in the'?

Lines 179, 360: 'high resolution' -> 'fine-resolution'

Table 1: url -> URL

Line 196: 'be' -> 'being'

Line 205:

Lines 217-232: Please also include some literature to prove that these involved sites are spatially representative at km scales or have been widely used in SSM validation.

Line 224: '2014)'

Line 262: is the '10-cm-depth' different from the '0-10 cm' like you mentioned in Line 229?

Line 285: Do you mean that one set of coefficients a-d will be used for all pixels of the whole country on t1?

Lines 309-311: I agree that STDF is enough for the accuracy requirement of soil moisture estimation. However, this explanation here is weird because the atmosphere does have interactions with the surface at cloudy-sky: cloudy conditions may also indicate it is raining or the atmosphere is wet. Such LST and ET disturbance signals, which can be captured by PM-based LST but not by STDF, will impact the

soil moisture. In other words, the atmospheric condition cannot be simply separated by using such an explanation.

Line 326: One or two sentences for briefly summarizing the downscaling methodology in Song et al. 2021 are necessary.

Line 329: SEE, "soil evaporative efficiency"

Line 346: "All pixels were utilized within … centered at … " would be better

Line 369: can you explain what "spatial averaging disaggregation" is

Line 417: why the bias caused by heterogeneity is negative?

Line 431: why RMSD_diff is important and focused? Maybe both clear-sky and cloudy-sky LSTs have higher uncertainty at some locations but the difference is small.

Fig 3: the absolute accuracy numbers of Fig 3(a) and (b) are better to be listed in the figure

Line 436: I feel 1.9 K is not small, and the RMSD difference can be ~70% of the clear-sky LST absolute accuracy [*Xu and Cheng*, 2021; *Zhang et al.*, 2021], especially for the nighttime LST. The word 'only' is too strong.

Fig 5, Line 663: please unify the ubRMSD or ubRMSE in the context.

**Citations**

Xu, S., and J. Cheng (2021), A new land surface temperature fusion strategy based on cumulative distribution function matching and multiresolution Kalman filtering, *Remote Sensing of Environment*, *254*, 112256.
Zhang, X., J. Zhou, S. Liang, and D. Wang (2021), A practical reanalysis data and thermal infrared remote sensing data merging (RTM) method for reconstruction of a 1-km all-weather land surface temperature, *Remote Sensing of Environment*, *260*, 112437.

---

## Author Comment (AC1)

**Author Response to RC1**

**Journal:** ESSD

**Title:** A 1-km daily surface soil moisture dataset of enhanced coverage under all-weather conditions over China in 2003-2019

**Author(s):** Peilin Song et al.

**MS No.:** essd-2021-428

**MS Type:** Data description paper

**General Comments:**

*"This paper proposes fine-resolution surface soil moisture (SSM) data over China. The significance and potential impact are clear, and the novelty and results are promising. However, a major revision is needed to address my concerns."*

**Response:**

All authors greatly appreciate you for your constructive comments that have helped improve our paper. We have paid great attentions on each bullet pointed out by you and have modified our paper carefully based on your comments. Please see the following responses to your specific comments and technical comments.

**Response to specific comments**

1) *"Line 236: About the LST validation:*

*Aqua nighttime passing time can be 1-hour away from 1:30 am LT. Even nighttime LST does have small variations than daytime LST, can you find some sites with minute-level observations in China to prove that using ground observations at 2:00 am introduces little uncertainty to the validation results?*

*Besides, the 0-cm ground temperature is different from LST physically, especially over vegetated areas, where SSM estimation by LST may have considerable uncertainty. Over these places, LST is closer to vegetation canopy temperature (air temperature).*

*To address my two concerns above, I would recommend including a brief test in the discussion by using site-measured LSTs that are computed by surface upward longwave radiation and BBE, and it would be convincing to include sites over various land cover types."*

**Response:**

Thanks for your suggestion. We have added a new Section 4.2 to discuss this question as well as your Questions 3 and 5 below. For the current discussion, please see Lines 669-686 for details. Basically, we cannot use flux towers to substitute validation data derived at meteorological stations because the spatial density and temporal coverage of the former dataset are not adequately high to represent the entire China. However, following your suggestion, we have implemented the test to address these two concerns in the added Section Appendix C.

In Appendix C, we selected 4 extra flux towers where long radiation observations are publicly available for a comparison with our 0-cm ground temperature, based on your suggestion. The minute-level LST of these towers between 1:00-2:00 A.M. are stable and consistent with the night-time 0-cm ground temperature at meteorological stations.

But one point we need to stress is that the meteorological sites are all located "under open environmental conditions with relatively lower fraction of tall trees and water bodies"(see Lines 677-679 in Section 4.2), according to the official regulation of the National Meteorological Administration of China. Also, it is difficult to find flux towers paired with meteorological stations over densely vegetated regions. Instead, the 4 towers are all located within grasslands across the country.

Besides, we have also re-checked the overpass time of MODIS LST product. The extreme time deviation from 1:30 A.M. can be about 15-20 minutes in our study period and region, not as large as one hour.

2) *"is there any evidence to prove the rationality of '7x7' and '-5$_{th}$ to 5$_{th}$'? "*
**Response:**

These two values have been actually determined as the optimal ones based on our test and evaluation against in-situ data from a collection of values. We have revised the paper by adding this description. Please see Lines 360-368.

3) *"Fig 3: Clear bias is still shown in filled LST results (Fig 3b) compared to the clear-sky validation (Fig 3a). Will it affect the SSM estimation when clear-sky (unbiased) and filled LSTs (biased) simultaneously exist in a spatial window using the 'universal triangle feature' or in SEE calculation?"*
**Response:**

Thanks for this comment. There is indeed inevitable influence for such clear-sky-to-cloud mixed windows when we intend for a dataset of quasi-complete coverage. Based on your question, we have added a brief discussion on this in the new added Section 4.2. Please see Lines 657-668.

In summary, such influence implies that the actual difference between SSM downscaling results at cloudy and at clear-sky conditions may be larger than "0.056 vol/vol VS 0.053 vol/vol". But overall, it should not affect the main features of the proposed product (e.g. the better performance of the STDF-derived LST in downscaling cloudy SSM compared to the bias-adjusted one). Also, such possible sacrifice for accuracy of clear-sky SSM in the clear-sky-to-cloud mixed windows can make the product accuracy more consistent between cloudy

and clear-sky conditions. This is beneficial to wider application of the product in future studies.

4.1) "*Fig 5: After readers notice the clear differences between two data at some locations (Fig 5a&b), they may want to know which data is more accurate.*
*In order to address this concern, you may need to focus on the sites over these regions, where the proposed data have considerable differences with SMAP-Sentinel (e.g. far northeastern, northern west, southern provinces near the sea), specifically and separately, rather than just over entire China (Line 499).*"
**Response:**

  We had actually carried out such analysis which is consistent with your suggestion. In this regard, we produced a map for demonstrating all available validation sites in terms of the direct ubRMSD difference (at each site) based on ubRMSD of SMAP-sentinel data minus that of the proposed product. From the map (see the Fig.1 below), however, we cannot find significantly different regional (e.g. between the northeast and the southwest) patterns of the "ubRMSD difference". As a consequence, we decided to maintain the current validation strategy for our paper. Detailed reasons are as follows:

  (1) First, from the map below we can see that the validation sites are not evenly distributed across the country, especially considering the much smaller number of sites in the southwestern part. This makes it difficult to make a fair comparison for different sub-regions.

  (2) Second, it is important to notice that validation of remote sensing soil moisture based on site measurements actually evaluates the similarity of the "trends" in both the spatial and the temporal dimensions between remote sensing and in situ data. But for the sub-regional validation, we can only evaluate the site-based temporal trend or spatial trend at a much smaller spatio scale but have to abandon the national-scale spatial trend which is especially important. This indicates that the overall validation across the country can be a more comprehensive and more fair validation strategy.

  (3) The main object of our paper is to develop a product of higher temporal resolution, higher coverage and higher accuracy than current data (SMAP-Sentinel). As with comparing the detailed qualities of different data products in different sub-regions, the map (in Fig.1 below) indicates that the inconsistent performances of the products cannot be simply ascribed to their differences on geographical locations or climatic regions. In reality, as the basic theories, data inputs, mathematical algorithms, and uncertainty sources differ completely between SMAP-Sentinel combined and PM-optical-data combined downscaling frameworks, the complexity of this issue may be beyond the center topic of current study and need to be investigated specifically in the future.

[Figure]

Fig. 1 The spatial distribution for Difference of single-site-based ubRMSD of SMAP-sentinel data minus that of our proposed product (ubRMSD$_{sentinel-smap}$ - ubRMSD$_{proposed}$), corresponding to Fig.5 in the paper. Sites with samples less than 20 for one year are excluded.

4.2) *"Besides, SMAP shows a very good accuracy (Fig A1a) while the downscaled SMAP-Sentinel (Fig 5c) has large (nearly doubled) ubRMSD. Can you explain why the accuracy is considerably decreased after downscaling?"*

**Response:**

According to the authors of the SMAP-Sentinel product (Das et al., 2019), uncertainty of this product includes that from its ancillary datasets, the optimization process on its model coefficients, as well as the increased speckle noise introduced when the spatial resolution of Sentinel-1 data is enhanced from 9 km to 1 km. Therefore, the authors of Das et al. (2019) comment that there is "tradeoff between adding spatial resolution with C-band SAR data and noise-levels". This can explain why SMAP-Sentinel has larger ubRMSD than SMAP data. This result is also supported by another previous evaluation study (Mohammad et al., 2018).

On the other hand, we understand that you may have concern on the result that the SMAP-Sentinel based ubRMSD is nearly doubled after downscaled. However, it is important to notice that the analyses in Appendix A and in Fig.5 are not based on the same numbers of validation sites. In Appendix A we only employed quite a small portion of the sites in only 53 microwave 36-km grids because only these sites have the qualified distribution density for representing the microwave grids. As these sites are mostly distributed in plain regions (see Fig. A1), there is a chance to further enlarge its performance difference with the SMAP-Sentinel based result because the latter is evaluated based on a much larger number of validation sites. As a conclusion, we can compare the performance difference between SMAP-Sentinel and our proposed data in Fig.5 as they are based on the equivalent sampling size, whilst it is more or less not fair to quantitatively analyze the decreased ubRMSD of

SMAP-Sentinel data against that of SMAP data between Fig.5 and Fig.A1.

5)*"Appendix B: It's strange that filled LST with considerable bias (-1.7 K) can achieve better SSM accuracy (0.058 vol/vol) than the SSM (0.064 vol/vol) from more accurate/realistic cloudy-sky LST in Fig. A3, and such accuracy difference is even larger than its difference with the clear-sky SSM (0.053 vol/vol, LST is unbiased). If that is the case, the logic behind it is that SSM is not sensitive to the LST, which is not right.*

*Besides, the LST bias explanation in Lines 309-311 is not convincing: if the filled LST has clear bias compared to site observations, it only means it cannot reflect the realistic surface condition."*

**Response:**
We basically accept your comment that Lines 309-311 is not convincing enough. Now we have modified and moved these sentences to Lines 634-668 in the new added Section 4.2 as a better and more open discussion. In brief, the STDF-derived LST under cloud with clear bias may not be suitable for all cloudy conditions, especially we agree with you that it is not suitable for rainy cloud. However, we argue that it can explain at least a substantial part of the non-rainy cloud condition. For the bias-adjusted cloud gap-filled LST, although it is better in reflecting the realistic surface condition, such mechanical relationships among cloud, LST and SSM can be beyond what has been described by the UTFS theory which was originally proposed for clear sky only (see the Fig.2 below for illustration).

[Figure]

Fig.2 Illustration of the UTFS theory under clear sky

In our revised discussion, therefore, the higher ubRMSD of STDF-derived LST compared to real clear-sky data (0.056 vol/vol VS 0.053 vol/vol) suggests such a gap-filling strategy (based on STDF alone) is not 100% perfect (especially for rainy weather which is the most difficult for the entire community of land surface remote sensing) and further improvements

are encouraged, whilst the even higher ubRMSD of non-bias or bias-adjusted LST under cloud (0.064 vol/vol) suggests that the STDF-derived LST is at least a better alternative compared to its bias-adjusted counterpart.

Meanwhile, we also need to stress that the results in Appendix B do not indicate "SSM is not sensitive to LST", because if it is not sensitive, all three groups of SSM should not have difference in their validation performance. The results just indicate the difference of "LST-SSM" interaction mechanisms between clear-sky and cloudy conditions.

**Response to technical comments**

| Comments | Response |
|---|---|
| Lines 50-56: references are necessary for the background knowledge introduction, especially for the potential application examples | Accepted. See the revised beginning of the Introduction. (Lines 54-58) |
| Line 87: "universal triangle feature (UTF)" or "triangle feature space (TFS)"? | Accepted. We revised and unified the term as "universal triangle feature space (UTFS)" through the text. |
| Line 91: please define the acronym UCLA | Done as suggested. (Line 93) |
| Line 109: ',' should be removed | Done as suggested. |
| Lines 78-80, 112-113: references, please. | For Lines 78-80, we have accepted your advice and added references. For Line 112-113, "the above-mentioned optical/infrared-data-based downscaling methods" have had their references listed above when each method was firstly described. So there is no need to repeat here. |
| Line 117: 'whilst .. even inferior' is not appropriate here. There is no such logic in the context unless you mean 'UTF-based methods are found even inferior to the DISPATCH in a typical humid region' | We have altered it to another formulation that weakens such a logic. See Lines 120-122 ("*As far as the UTFS-based method is concerned, a poorer performance was obtained compared to the DISPATCH in a typical water-limited region in North America*"). The main idea we intend to convey is that universality for both of the methods is not perfect enough currently. |
| Lines 124: the objectives you mentioned here are more like broad impacts or potential significance while the objective of a study should be specific. | Accepted. We have changed the term as "potential significance", because the objective has been described just in the current paragraph (to produce the data product, Lines 123-127) |
| Line 144: 'after' or 'in the'? | Accepted. Changed to "in the". |
| Lines 179, 360: 'high resolution' -> 'fine- | Done as suggested. |

| | |
|---|---|
| resolution' | |
| Table 1: url -> URL | Done as suggested. |
| Line 196: 'be' -> 'being' | Done as suggested. |
| Line 205: 'the' | Accepted and removed. |
| Lines 217-232: Please also include some literature to prove that these involved sites are spatially representative at km scales or have been widely used in SSM validation. | Accepted. Please see Lines 231-232 for the added literatures showing it has been widely used. |
| Line 224: '2014)' | Done as suggested. |
| Line 262: is the '10-cm-depth' different from the '0-10 cm' like you mentioned in Line 229? | Accepted. They are the same in effect. We have revised the former as "0-10 cm". |
| Line 285: Do you mean that one set of coefficients a-d will be used for all pixels of the whole country on t1? | Yes. We have actually tested different solutions including sub-region-based coefficients and the common set of coefficients for all of the country. The data outcome based the common set of coefficients for a certain date has the best quality (obtaining both high accuracy and high coverage). |
| Lines 309-311: I agree that STDF is enough for the accuracy requirement of soil moisture estimation. However, this explanation here is weird because the atmosphere does have interactions with the surface at cloudy-sky: cloudy conditions may also indicate it is raining or the atmosphere is wet. Such LST and ET disturbance signals, which can be captured by PM-based LST but not by STDF, will impact the soil moisture. In other words, the atmospheric condition cannot be simply separated by using such an explanation. | We appreciate your agreement on our methodology.

For the discussion, you can see our detailed response to your Specific Comment 5 above.

Basically, we agree with you that atmospheric condition can bring extra uncertainty when we use STDF-derived LST as input, and we have better discussed it in the new added Section 4.2. However, this uncertainty can be smaller compared to using the bias-adjusted LST which is also not suitable for the downscaling theory we base as the theory was actually developed for clear-sky condition. |
| Line 326: One or two sentences for briefly summarizing the downscaling methodology in Song et al. 2021 are necessary. | Accepted and added. Please see Lines 334-338 |
| Line 329: SEE, "soil evaporative efficiency" | Done as suggested. |
| Line 346: "All pixels were utilized within … centered at … " would be better | Done as suggested. |
| Line 369: can you explain what "spatial averaging disaggregation" is | Sorry for this mistake. We have revised it as "spatial averaging operator for…" |
| Line 417: why the bias caused by heterogeneity is negative? | We actually mean the effect is "not beneficial", but not mean it has a "negative-sign bias". Now we have |

| | |
|---|---|
| | revised it to "disadvantageous effects" (Line 438) |
| Line 431: why RMSD_diff is important and focused? Maybe both clear-sky and cloudy-sky LSTs have higher uncertainty at some locations but the difference is small. | The main technical issues to tackle for generating this SSM product include gap-filling of LST under cloud, but not include retrieval of LST under clear sky. In other words, our work relies on the general accuracy of the existing LST product under clear sky (MODIS 1-km LST), which has been generally evaluated by Fig.3-(a) for the overall situation. As the primary purpose is to obtain LST of complete-coverage with consistent accuracy for all-weather conditions, the RMSD_diff is most important compared to other metrics. Based on the above concern, we made an extra site-based analysis for RMSD_diff in Fig.3-(c), while for the absolute RMSD values, the all-site analyses in Fig.3-(a) and –(b) are sufficient. |
| Fig 3: the absolute accuracy numbers of Fig 3(a) and (b) are better to be listed in the figure | Done as suggested. |
| Line 436: I feel 1.9 K is not small, and the RMSD difference can be ~70% of the clear-sky LST absolute accuracy [*Xu and Cheng*, 2021; *Zhang et al.*, 2021], especially for the nighttime LST. The word 'only' is too strong. | Accepted. We removed the word "only". We also changed the following sentence from "small uncertainty" to "uncertainty is not very significant". (Lines 458-460) |
| Fig 5, Line 663: please unify the ubRMSD or ubRMSE in the context. | Done as suggested. |

**Reference**

Das, N. N., Entekhabi, D., Dunbar, R. S., Chaubell, M. J., Colliander, A., Yueh, S., . . . Thibeault, M.: The SMAP and Copernicus Sentinel 1A/B microwave active-passive high resolution surface soil moisture product, Remote Sens. Environ., 233, 111380, https://doi.org/10.1016/j.rse.2019.111380, 2019.

Mohammad, E. H., Nicolas, B., Mehrez, Z., Nemesio, R. F., Jean, W., Amen, A. Y., . . . Jean-Christophe, C.: Evaluation of SMOS, SMAP, ASCAT and Sentinel-1 Soil Moisture Products at Sites in Southwestern France, Remote Sens., 10, 569, 2018.

---

## Author Comment (AC2)

**Author Response to RC2**

**Journal:** ESSD

**Title:** A 1-km daily surface soil moisture dataset of enhanced coverage under all-weather conditions over China in 2003-2019

**Author(s):** Peilin Song et al.

**MS No.:** essd-2021-428

**MS Type:** Data description paper

**General Comments:**

*"The authors present a downscaled soil moisture product, which combines the advantages of a 36-km resolution passive microwave remote sensing product with a 1-km resolution MODIS LST product. Such high-resolution soil moisture is very important for agriculture and water resource management. The manuscript is generally well-organized, I suggest accepting it with considering the following revisions."*

**Response:**

All authors greatly appreciate you for your final decision with "accepting it with considering the following revisions". We have paid great attentions on each bullet pointed out by you and have modified our paper carefully based on your comments. Please see the following responses to your specific comments.

**Response to specific comments**

1.  The quality of the figures should be improved. Currently, some legends are too small to identify.

**Response:**

We have tried to improve the quality of some figures (Fig.3, Fig.5, Fig. 7). However, if there are still unclear legends in the figures, please let us know the specific points after this revision. Thank you.

2. Fig. 7. was not used in the main text.

**Response:**

We accept your comment and have mentioned Fig.7 as "another manner of illustrating Fig.6" in the revised version, above Fig.6 and Fig.7. Please see Lines 544-547 ("*The above inter-seasonal differences on data coverage are also reflected in Fig. 7 in another manner based on presenting the spatial distributions of number percentages of available dates in each three-month period*").

3. Also in Fig.7., it shows that the original PM SSM almost does not have any data in the winter season on the Tibetan Plateau. It is reasonable since, in the winter season, the soil is frozen and generally covered by snow, and then it is difficult for microwave remote sensing to identify soil moisture. However, as shown in this figure, the new 1-km downscaling product has some soil moisture data. How did it come? What did the soil moisture value during this season on the Tibetan Plateau mean? How about the accuracy of these downscaled SSM?

**Response:**

Thank you for reminding us on this problem. Our downscaling framework is actually consistent with your opinion on leaving out the invalid "winter pixels" (See our description in Lines 538-542). Unfortunately, we made a tiny technical mistake when calculating the statistics for Fig. 6 and Fig.7 last time. Now the bug has been fixed for both Fig. 6 and Fig. 7. Also, relevant texts have been revised (please see Line 535). In this revised version, null values have been assigned for all 1-km sub-pixels within the frozen or snow-covered passive microwave pixels (e.g. the microwave pixels characterized by null values on the Tibetan Plateau in winter and early spring).

4. It is recommended to draw some time series of the soil moisture products, the new one, the original one, and SMAP high resolution one, on several stations, to demonstrate the advantages of this daily 1-km product.

**Response:**

Thanks for your advice. We had actually investigated the time series at some of the stations when we designed this study. After careful investigation, however, we found it is rather difficult to use time series data at only a few stations to highlight our conclusions that have been drawn based on nation-wide research. We do not wish to have an impression of cherry picking. Therefore, we finally decided not to present any of them in the paper:

(1) In our study we have more than 2000 validation sites across the country in total. The time series patterns for the downscaled SSM and the station benchmarks are rather different from site to site. It's very difficult to find one or two sites where the relative performances

among soil moisture time series of different data sources are typical and representative of their background climate regions at the provincial or large-basin levels. Moreover, we believe the complicated influential factors behind soil moisture seasonal time series of different eco-regions have to be investigated specifically in our subsequent studies. For our current study case, the overall validation performance (see Fig.4) is more important than time series demonstration.

(2)  The SMAP-sentinel high resolution data has a much poorer temporal frequency (for some locations even lower than 12 days), as a consequence of which, the true shape of its time series might be arbitrarily interpreted. Therefore, it is difficult to fairly compare its time series with that of other daily-scale datasets through visual inspection.

---

## Referee Report (RR1)

**General Comments:**

After the revision, the manuscript has resolved some of my concerns. However, I found some severe issues from the proposed dataset, which need to be addressed. A careful revision is still necessary.

1. Feedback towards the response letter

1) The authors emphasized that the proposed dataset aims to capture spatiotemporal trends national wide, and regional uncertainty analysis in Fig. 5, or temporal analysis requested from reviewer #2, are not focused. Site distribution could be a limitation but won't affect the analysis.

Can you provide temporal analysis at typical regions, and spatial uncertainty analysis (as in Fig. 5), by comparing the data with SMAP? Temporal variation comparison between regional averages of two datasets can prevent the 'cherry picking' and resolution mismatch issue. (Maybe more datasets are better, to see if this proposed data is not consistent with the majority.) Even though SMAP has a coarse resolution, it still has relatively good accuracy, spatiotemporal continuity, and reliability from passive microwave observations. If the study focuses on national scale analysis, coarse resolution won't be a problem.

What I would like to point out here is that, the advantage of a dataset with a high spatiotemporal resolution is to do regional analyses, not a national scale. Therefore, I would still recommend the authors provide such detailed regional and temporal assessment.

Additionally, considering that the model parameters are obtained mainly based on spatial information national wide, accuracy stability at the time dimension is very important.

2) SM and radiative temperature have strong interactions during the daytime due to ET and energy partitioning, unlike other "triangle method"-based studies, why does the proposed method only used nighttime LST, when the whole energy partitioning process is very weak? (even Fig. 2 of the response letter shows the daytime relationship between LST and vegetation cover)

2. Issues with the data

After I downloaded the proposed dataset, I found several issues, especially focusing on regional levels, and that is why I start to suspect its ability to work on regional studies. Taking Day 2008053 as an example:

1) mosaic issue/spatial discontinuity

[Figure]

Clearly severe mosaic patterns are illustrated. Such an issue is also obvious in northeastern China on this day. In fact, I can find similar mosaic patterns in most images and places, such as a day (Day 2008210) with good spatial completeness at central south of China:

[Figure]

Pixels at connection regions among mosaics will have large uncertainties. Such mosaic issue is not reflected in Fig. 5. This problem should be focused on because it will affect the feasibility in regional studies.

2) Besides, there are lots of randomly scattered high SM values (case Day 2008053), which seems not correct in the dry region:

[Figure]

3) There are no coordinate, projection, or geolocation information in the dataset, causing it hard to be used. Moreover, I would like to recommend the authors include 'QC' band in the dataset in the future version, based on comprehensive uncertainty analysis and available input data.

**Technical comments**

1. Line 54: blank missing before '('

---

## Author Response (AR2)

Dear Editor and Reviewers,

Thank you for your attention. We appreciate your earnest work including comments and suggestions concerning our manuscript. Based on the comments, we have made careful modifications on the original manuscript. As required by this journal, the responses to the referees have been structured as follows: (1) comments from Referees and corresponding author's response, and (2) author's changes in manuscript. Therefore, we have responded to the reviewers in the sequence: (1) the original comments in black and our point-by-point responses in blue, and (2) our revised manuscript highlighted using "track change". All line numbers in the responses are made with respect to this "track change" version of the manuscript.

The details of the response to the referee and the corresponding revised manuscript are shown in the following section. We hope that the revised manuscript at this stage could be qualified for potential publication, and we look forward to hearing from you soon.

Yours sincerely,

Prof. Yongqiang Zhang; Dr. Peilin Song

Key Laboratory of Water Cycle and Related Land Surface Processes, Institute of Geographic Sciences and Natural Resources Research, The Chinese Academy of Sciences, Beijing 100101, China

Emails: zhangyq@igsnrr.ac.cn; songpl@igsnrr.ac.cn

**Author Response to Referee-report-2**

**Journal:** ESSD

**Title:** A 1-km daily surface soil moisture dataset of enhanced coverage under all-weather conditions over China in 2003-2019

**Author(s):** Peilin Song et al.

**MS No.:** essd-2021-428

**MS Type:** Data description paper

**Response to specific comments**

*1. Feedback towards the response letter*
*1) The authors emphasized that the proposed dataset aims to capture spatiotemporal trends national wide, and regional uncertainty analysis in Fig. 5, or temporal analysis requested from reviewer #2, are not focused. Site distribution could be a limitation but won't affect the analysis.*
*Can you provide temporal analysis at typical regions, and spatial uncertainty analysis (as in Fig. 5), by comparing the data with SMAP? Temporal variation comparison between regional averages of two datasets can prevent the 'cherry picking' and resolution mismatch issue. (Maybe more datasets are better, to see if this proposed data is not consistent with the majority.) Even though SMAP has a coarse resolution, it still has relatively good accuracy, spatiotemporal continuity, and reliability from passive microwave observations. If the study focuses on national scale analysis, coarse resolution won't be a problem.*
*What I would like to point out here is that, the advantage of a dataset with a high spatiotemporal resolution is to do regional analyses, not a national scale. Therefore, I would still recommend the authors provide such detailed regional and temporal assessment. Additionally, considering that the model parameters are obtained mainly based on spatial information national wide, accuracy stability at the time dimension is very important.*
Response:
    We have accepted your suggestions and added temporal analysis of our developed product against SMAP 36-km SSM by dividing China into six geographic regions. Please see Lines 512-528 as well as the new added Fig.5 for details. Descriptions about the study area in Section 2.1.3 is also refined.
    We also have added spatial uncertainty analysis for both our developed data and the SMAP-sentinel data against SMAP 36-km SSM. The results are shown in Fig.6 (Just the original Fig.5)-(c) and –(d). Please see relevant analyses in Lines 546-556. Clearly, the advantage of our developed product over the existing SMAP-Sentinel combined product is found mainly in the south-western part of the country with increased topographic effects.

*2) SM and radiative temperature have strong interactions during the daytime due to ET and energy partitioning, unlike other "triangle method"-based studies, why does the proposed method only used nighttime LST, when the whole energy partitioning process is very weak? (even Fig. 2 of the response letter shows the daytime relationship between LST and vegetation cover)*

**Response:**

Thanks for your questions. We'd like to explain the reason for using nighttime LST from the following perspectives.

First, we found that Dr.Merlin's team indeed prefer using daytime LST for downscaling the early morning SMAP or SMOS SSM in their studies. However, other study (Piles et al., 2014) reports close performances between using daytime and nighttime LST. Such inconsistent results can at least lead to a conclusion that whether the daytime LST outperforms the nighttime LST may depend on the specific case. From the point of view of physical mechanisms, LST influences SSM through the intermediate parameter of evaporative efficiency (i.e. EE, $=\frac{LST_{max}-LST_{modis}}{LST_{max}-LST_{min}}$). We herein made a simple experiment, based on the

Aqua MODIS daytime and nightime LST images in China on June 30, 2018. We defined the $LST_{max}$ and $LST_{min}$ as the maximum and minimum LST of each 36-km pixel, and then calculated EE for each 1-km LST. The day-time and night-time comparisons are shown in the following Fig.1. Fig.1-(a) shows that the LST variation in the night time is weaker than in the day time. From Fig.1-(b), however, the variation range for day-time and night-time EE is much closer. This indicates that influences of weaker energy partitioning process in the night time on EE estimates (and SSM downscaling performance) is not so strong.

[Figure]

Fig.1 An experiment illustrating difference between day-/night- time LST (a) and day-/night-time EE (b), using Aqua MODIS data on June 30, 2018.

Second, as we have responded in the last round of revision, LST cloud-gap-filling results at night have higher reliability than in the day time. Also, consistency between satellite and ground-based temperature measurements is better at night than in the day time. Therefore, using night-time LST can generate more consistent SSM estimates for all-weather conditions, and can better implement validation of the SSM product based on ground measurements.

Besides, as night-time LST are synchronously observed with AMSR-2 TB, this may avoid influence from rapid soil moisture change during the time lag between day-time and night-time observations.

Based on all above, we believe using night-time LST is a better choice for downscaling PM SSM for our study case.

*2. Issues with the data*

*After I downloaded the proposed dataset, I found several issues, especially focusing on regional levels, and that is why I start to suspect its ability to work on regional studies. Taking Day 2008053 as an example:*

1) *mosaic issue/spatial discontinuity. Clearly severe mosaic patterns are illustrated. Such an issue is also obvious in northeastern China on this day. In fact, I can find similar mosaic patterns in most images and places, such as a day (Day 2008210) with good spatial completeness at central south of China: pixels at connection regions among mosaics will have large uncertainties. Such mosaic issue is not reflected in Fig. 5. This problem should be focused on because it will affect the feasibility in regional studies.*

   **Response:**

   Thanks for your comments to our data product. We admit that the mosaic issue is one that we cannot completely solve based on current downscaling frameworks. Please see Equation-5 in Section 2.2.2. In this downscaling equation, the 1-km SSM is the sum of the 36-km PM SSM plus 1-st and even higher orders of derivatives of a specific model function. This means, the spatial texture of the 36-km PM SSM is inherently contained in the finally downscaled dataset. For cases with drastically varied texture on neighboring PM pixels (for such cases, one of the PM SSM retrievals may have relatively lower reliability), the mosaic issue is raised for the downscaled SSM.

   Actually, this problem exists for all mapping results shown in existing studies like (Molero et al., 2016; Stefan et al., 2020; Peng et al., 2016). Since our study is conducted at a much large spatial area, the problem can be even more obvious especially for special land cover conditions that may influence the accuracy of PM SSM (e.g. in the Qinghai-Tibet Plateau with complicated topography, melt snow or partially frozen soils that cannot been completely screened out by the PM product flag in winter).

   As mentioned in Section 2.2.2, we have used a parameter of 'spatial square window (ws)' in Equation-(3) to decline this negative effect to the best of our ability. But overall, we believe the fundamental reason for this problem is related to the quality of the PM SSM, as well as other uncertainty sources discussed in Section 4.1 and 4.2. Therefore, completely solving this problem requires a series of improvement in future studies.

   We have added a paragraph to specifically discuss this issue. Please see Lines 767-781 in Section 4.3.

2) *Besides, there are lots of randomly scattered high SM values (case Day 2008053), which seems not correct in the dry region:*

   **Response:**

The algorithm is consistent for all pixels including the scattered ones. Although these scattered high SSM values may contain higher uncertainties, possibility does exist that they represent a situation of partially melt snow or ice-water mixture in the northwestern China in winter. As our ground observation data cannot evaluate such situations accurately in the current time, we finally decided to preserve such pixels and leave them for "further investigation through field survey or experiments". Please see our added discussion on this issue in Lines 775-781 in Section 4.3.

*3)    There are no coordinate, projection, or geolocation information in the dataset, causing it hard to be used. Moreover, I would like to recommend the authors include 'QC' band in the dataset in the future version, based on comprehensive uncertainty analysis and available input data.*

**Response:**

The projection and geo-transform information is stored in the metadata (Global_Attributes) of each HDF5 files. Such a file structure is learned from that of NASA MODIS datasets. We suggest you to examine it through a professional HDF5-reading software like Beam-VISAT. We are contacting the manager of the TPDC (http://data.tpdc.ac.cn/ ) web database for uploading an updated user guideline on this information. However, this may take extra time before the updated guideline is available.

We appreciate your suggestion on the 'QC' band. We initially planned to create this band, but has finally decided to delay it to the future version, before which we may need a more comprehensive judgement on the data product based on feedback from community users.

**Technical comments**

1. Line 54: blank missing before '('

   **Response:**

   Revised already.

**Reference**

[revised manuscript text omitted]